# LEARNING TO OPTIMIZE QUASI-NEWTON METHODS

## ABSTRACT

We introduce a novel machine learning optimizer called LODO, which online meta-learns an implicit inverse Hessian of the loss as a subroutine of quasi-Newton optimization. Our optimizer merges Learning to Optimize (L2O) techniques with quasi-Newton methods to learn neural representations of symmetric matrix vector products, which are more flexible than those in other quasi-Newton methods. Unlike other L2O methods, ours does not require any meta-training on a training task distribution, and instead learns to optimize *on the fly* while optimizing on the test task, adapting to the local characteristics of the loss landscape while traversing it. Theoretically, we show that our optimizer approximates the inverse Hessian in noisy loss landscapes and is capable of representing a wide range of inverse Hessians. We experimentally verify our algorithm's performance in the presence of noise, and show that simpler alternatives for representing the inverse Hessians worsen performance. Lastly, we use our optimizer to train a semi-realistic deep neural network with 95k parameters, and obtain competitive results against standard neural network optimizers.

## 1 INTRODUCTION

Many optimization algorithms like stochastic gradient descent (SGD) (Rosenblatt, 1958), and Adam (Kingma & Ba, 2014) have been widespread and successful in the rapid training of deep neural networks. (Sun et al., 2019) Fundamentally, this is a problem of minimizing a loss which is a function of a large vector containing the weights of the network. The time it takes to optimize a neural network is a bottleneck in machine learning, so the more quickly a network can be trained, the more computational resources are saved, and therefore researchers have devoted great effort into creating new, faster optimizers. (Jain & Kar, 2017; Metz et al., 2020; Bernstein et al., 2020; Martens & Grosse, 2015a)

We present a novel algorithm drawing from the field of learning to optimize (L2O) spearheaded by (Li & Malik, 2016) and (Andrychowicz et al., 2016). Namely, we use a meta-optimizer to online learn an implicit representation of the local inverse Hessian, which is used in a quasi-Newton method, without any L2O meta training time on a training task distribution. Unlike other L2O algorithms which learn to optimize *before* optimization (Chen et al., 2021), our algorithm Learns to Optimize *During* Optimization (LODO). We intend for LODO to be trained from scratch for each use case and then discarded. This way, LODO learns local features of the loss landscape at a specific point in training for a specific task, instead of only characteristics shared throughout training trajectories for a set of training tasks. Our work targets the Hessian, which varies with both the task and the point along the trajectory. Our use of linear neural networks is what imports the efficiency of the Newton method to our algorithm, while our use of a meta-optimizer like in L2O is what allows us to learn more powerful and general parameterizations of optimizers.

Our contributions are as follows. We show theoretically and experimentally that a simplified version of LODO correctly learns the inverse Hessian in a stochastic convex setting. We show theoretically that LODO's inverse Hessian representation is highly expressive, and experimentally that simpler alternatives perform worse. We finally demonstrate the use of LODO in a semi-realistic vision task. This paper serves as a stepping stone in the development of meta-training-free online L2O.

The remainder of this paper is structured as follows. Section 2 discusses relevant background and contributions in optimization and L2O. Section 3 shows how LODO works. Section 4 provides

theoretical justification for our design of LODO. Sections 5 and 5.2 show experiments which explore what makes LODO work and why. Section 6 discusses and summarize our findings.

## 2 RELATED WORK

Research into the construction of faster optimizers has mostly fallen under two branches of work. The older branch attempts to endow SGD with adaptive capabilities, often through modifications involving calculation of the first and/or second moments of the gradient (mean and variance) using exponential moving averages (EMAs). RMSprop (Hinton et al., 2012) and Adam use the variance to normalize the step size and the mean to induce momentum. LARS (You et al., 2017) and Yogi (Zaheer et al., 2018) modify both modify the variance calculation, but for different reasons: to normalize layer-wise, and to control increases in effective learning rate in a slower manner, respectively.

Some of these methods such as the Newton method and natural gradient descent (Martens & Grosse, 2015b; George et al., 2018) precondition the step with adaptive estimates of the inverse Hessian and the inverse Fisher information matrices, respectively. The Newton method converges quickly but is vulnerable to gradient noise and impractical to implement due to the resources spent in calculating and/or inverting the high dimensional Hessian. Many researchers have developed approximations— called quasi-Newton methods—which reduce the Newton method's time and memory complexity, such as L-BFGS (Nocedal & Wright, 1999) and variants (Schraudolph et al., 2007; Parker-Holder et al., 2020; Goldfarb et al., 2020; Park & Oliva, 2019) better suited to the stochasticity and structure present in machine learning. The most related methods to our work are hypergradient methods, which online learn low-rank (Moskovitz et al., 2019), diagonal (Amid et al., 2022; Baydin et al., 2017), or Kronecker-factorized (Bae et al., 2022) preconditioner matrices to transform the gradient when choosing the step. We improve on these methods by using a more expressive class of preconditioners.

More recently, a subfield of meta-learning known as learning to optimize (L2O) has shown that deep networks can themselves be trained to perform optimization, at a speed which exceeds that of popular traditional optimizers. The aim of this effort is to leverage deep neural networks to learn faster optimizers, in hopes of further accelerating training procedures for other deep neural networks. Li & Malik (2016; 2017) and Andrychowicz et al. (2016) were among the first to successfully use backpropagation to train neural networks to map gradients to steps. Since then, many other variations of this idea have successfully produced optimizers exceeding the speed of common optimizers for narrow ranges of machine learning models (Metz et al., 2018), though theoretical analysis of these learned optimizers tends to be difficult and scarce. A major goal of L2O research is to learn a single optimizer which can generalize to be able to train a wide variety of machine learning models with speed. (Lv et al., 2017)

Two issues also prevent L2O optimizers from being rapidly developed experimentally. Firstly, a carefully chosen "task distribution" for the optimizer to practice on is required for the meta-learning of the L2O optimizer, playing the role analogous to the "dataset". These tasks are difficult to curate because the issue of generalization error applies; we want the test task to be similar to the task distribution. Secondly, this meta-learning of the L2O optimizer is prohibitively costly, in that it involves nested training loops, where the inner loop takes a large amount of time and memory to evaluate and backpropagate through (Metz et al., 2019). Altogether, the choice of task distribution and lengthy meta-training has been a necessary burden in L2O, and we overcome these with LODO.

## 3 HOW LODO WORKS

In a quasi-Newton method, the approximate solution $\boldsymbol{x}_t \in \mathbb{R}^n$ is refined by $\boldsymbol{x}_{t+1} = \boldsymbol{x}_t - \tilde{\alpha} \boldsymbol{G}_t \boldsymbol{g}_t$ for some learning rate $\tilde{\alpha} > 0$, where $\boldsymbol{G}_t \approx (\nabla_{\boldsymbol{x}_t}^2 f(\boldsymbol{x}_t))^{-1} \in \mathbb{R}^{n \times n}$ is some approximation of the inverse Hessian and $\boldsymbol{g}_t = \nabla_{\boldsymbol{x}_t} f(\boldsymbol{x}_t) \in \mathbb{R}^n$ is the gradient computed by backpropagation through the task $f$. $\tilde{\alpha} = 1$ produces the exact solution if $f$ is quadratic, so we set $\tilde{\alpha} = 1$. Our algorithm approximates the inverse Hessian using a matrix $\boldsymbol{G}(\boldsymbol{\theta}_t) \in \mathbb{R}^{n \times n}$ parameterized by a vector $\boldsymbol{\theta}_t$ of weights learned over time $t$, described later in this section. After every step $t \leftarrow t + 1$ using the formula $\boldsymbol{x}_{t+1} = \boldsymbol{x}_t - \boldsymbol{G}(\boldsymbol{\theta}_t) \boldsymbol{g}_t$, the loss $f(\boldsymbol{x}_{t+1})$ is computed. Then the new gradient $\nabla_{\boldsymbol{x}_{t+1}} f(\boldsymbol{x}_{t+1})$ in $\boldsymbol{x}_{t+1}$ is computed through backpropagation as usual, but we continue backpropagation into the step-choosing process until we find the "hypergradient" $\nabla_{\boldsymbol{\theta}_t} f(\boldsymbol{x}_{t+1})$ in the optimizer weights $\boldsymbol{\theta}_t$,

---

**Algorithm 1** Learning to Optimize During Optimization (LODO)

---

**Require:** $f : \mathbb{R}^n \to \mathbb{R}$, $\boldsymbol{x}_0 \in \mathbb{R}^n$: Function to minimize, initialization.
**Require:** $\alpha \in \mathbb{R}$: Meta-learning rate (default 0.001),
**Require:** $\alpha_0 \in \mathbb{R}$: Initial learning rate (default 1.0),
**Require:** $0 \le \beta < 1$: Momentum (default 0.9),
     $t \leftarrow 0$        ▷ Start time
     $\boldsymbol{\theta}_0 \leftarrow$ random initialization        ▷ Initialization for neural network.
     $\boldsymbol{m}_0 \leftarrow \boldsymbol{0}$        ▷ Initialize EMAs
     **while** not converged **do**
         $\boldsymbol{x}_{t+1} \leftarrow \boldsymbol{x}_t - \boldsymbol{G}(\boldsymbol{\theta}_t, \alpha_0)\boldsymbol{m}_t$    ▷ Pick step with inverse Hessian model $\boldsymbol{G}$ from Equation (1)
         $\ell_{t+1} \leftarrow f(\boldsymbol{x}_{t+1})$        ▷ Compute loss after step
         $\boldsymbol{m}_{t+1} \leftarrow \beta \boldsymbol{m}_t + (1-\beta)\nabla_{\boldsymbol{x}_{t+1}}\ell_{t+1}$        ▷ Update gradient EMAs
         update $\boldsymbol{\theta}_t \to \boldsymbol{\theta}_{t+1}$ using Adam with $\nabla_{\boldsymbol{\theta}_t}\ell_{t+1}$        ▷ Tune the model to choose a better step
         $t \leftarrow t + 1$        ▷ Increment time
     **end while**
     **return** $\boldsymbol{\theta}_t$

---

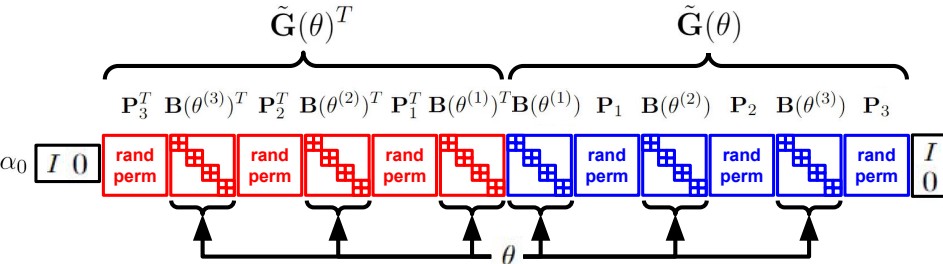

Figure 1: Visualization of LODO's matrix structure of $\boldsymbol{G}(\boldsymbol{\theta})$ from Equation (1) for approximation of the inverse Hessian. Reduced to a depth of 3, block size 2, and size $\tilde{n} = 8$ matrices for illustration.

which allows us to update them as well. Thus, $\boldsymbol{\theta}$ is trained such that the quasi-Newton method tries to minimize the loss upon taking a single step.

In the L2O interpretation, this would be equivalent to unrolling the inner loop optimization for only one step, causing severe truncation bias; the optimizer learns to greedily optimize within too short of a time horizon, thus suffering in the long term (Metz et al., 2019). As a countermeasure, we take inspiration from the Momentum modification to SGD, and replace $\boldsymbol{g}_t$ by $(1-\beta)\sum_{\tau=0}^{\infty}\beta^{t-\tau}\boldsymbol{g}_{\tau}$ before input into the step generation formula. This changes LODO's step generation formula to $\boldsymbol{x}_{t+1} = \boldsymbol{x}_t - (1-\beta)\boldsymbol{G}(\boldsymbol{\theta}_t)\sum_{\tau=0}^{\infty}\beta^{t-\tau}\boldsymbol{g}_{\tau}$ instead, for some decay parameter $\beta$. This results in LODO, summarized in Algorithm 1.

Our parameterization of $\boldsymbol{G}(\boldsymbol{\theta})$ separates our work from other hypergradient methods; ours is inspired by the FFT-style Efficient Unitary Neural Network (EUNN) (Jing et al., 2017) designed for parameter-efficient representations of unitary matrices. We replace the FFT matrices in the EUNN with fixed randomly chosen sparse matrices, and write $\boldsymbol{G}(\boldsymbol{\theta})$ to be a large product of matrices:

$$\boldsymbol{G}(\boldsymbol{\theta}) = \alpha_0 \begin{pmatrix} \boldsymbol{I} & \boldsymbol{0} \end{pmatrix} \tilde{G}(\boldsymbol{\theta})^T \tilde{G}(\boldsymbol{\theta}) \begin{pmatrix} \boldsymbol{I} \\ \boldsymbol{0} \end{pmatrix}; \quad \tilde{G}(\boldsymbol{\theta}) = \prod_{i=1}^{N} \boldsymbol{B}(\boldsymbol{\theta}^{(i)})\boldsymbol{P}_i \tag{1}$$

where $\alpha_0$ is a fixed initial learning rate, $\boldsymbol{P}_i$ are randomly selected permutation matrices, $\boldsymbol{B}(\boldsymbol{\theta}^{(i)})$ are block-diagonal matrices whose contents are listed by $\boldsymbol{\theta}^{(i)}$ where $\boldsymbol{\theta} = (\boldsymbol{\theta}^{(1)}, \ldots \boldsymbol{\theta}^{(N)})$, as illustrated in Figure 1. Every block is size $k \times k$ for some chosen $k$ (we use 4 for our setup), where the blocks' contents are listed by $\boldsymbol{\theta}$. The $\begin{pmatrix} \boldsymbol{I} & \boldsymbol{0} \end{pmatrix}$ and $\begin{pmatrix} \boldsymbol{I} & \boldsymbol{0} \end{pmatrix}^T$ matrices are $n \times \tilde{n}$ and $\tilde{n} \times n$ respectively, and all other matrices are $\tilde{n} \times \tilde{n}$, where $\tilde{n}$ is some chosen integer larger than $n$. For our setup, we choose $\tilde{n} = \lfloor 2n/k \rfloor k$. Though $\boldsymbol{G}$ depends on both $\alpha_0$ and $\boldsymbol{\theta}$, we omit the $\alpha_0$ in writing for brevity. By initializing each block matrix in $\boldsymbol{B}(\boldsymbol{\theta}^{(i)})$ to a random orthogonal matrix, we initialize $\boldsymbol{G}(\boldsymbol{\theta})$ to a multiple of the identity, despite the input information diffusing and mixing with itself inside the layers of the network. The initialization to the identity also allows us to make the depth $N$ large without worrying about issues with gradient magnitudes, and for our setup we choose $N = 16$.

The product $G(\boldsymbol{\theta})$ intentionally resembles the operation performed by a deep neural network with $N$ layers, millions of hidden neurons arranged in inverted bottleneck style, very sparse connections, and no activation functions. The random connections between neurons are intended to bring expander graph properties to the computation graph of the neural network, such that signals can diffuse and self-mix quickly without travelling long distances within the computation graph. Since we wanted LODO to be repelled by saddle points, we chose $G(\boldsymbol{\theta})$ to produce a step which has guaranteed positive dot product with the gradient, so we parameterized $G(\boldsymbol{\theta})$ in a way which guarantees it to be a positive-semidefinite symmetric matrix. Furthermore, since any positive-semidefinite symmetric matrix can be eigendecomposed into $\boldsymbol{U}\boldsymbol{D}\boldsymbol{U}^T = (\boldsymbol{U}\boldsymbol{D}^{1/2})(\boldsymbol{U}\boldsymbol{D}^{1/2})^T$ for unitary $\boldsymbol{U}$ and positive-semidefinite diagonal $\boldsymbol{D}$, we chose $G(\boldsymbol{\theta})$ to represent the product of a matrix with its transpose.

Since we expect the receptive field of output nodes to grow exponentially with depth, it takes $O(\log n)$ layers for all of the $n$ inputs to interact with each other, so we intend the depth $N$ to be some multiple of $\log n$. Applying permutations and block diagonal matrices both cost $O(n)$ time, so the inverse Hessian estimate $G(\boldsymbol{\theta})$ costs $O(n \log n)$ time. This cost is small if the task is to train an architecture with much more parameter reuse than $\log n$, such a CNN or transformer. The memory LODO needs is $O(n \log n)$ since we have $O(n \log n)$ learnable parameters, though other L2O optimizers which cost $O(n)$ may easily outstrip this cost via a large enough constant factor, for example by storing and processing a long history of past gradients.

## 4 THEORETICAL PROPERTIES OF LODO

In this section, we illustrate LODO's behavior through its theoretical properties. To make our analysis easier to show, this section only refers to LODO that is modified in the following ways: (*i*) We update $\boldsymbol{\theta}_t$ using SGD of learning rate $\alpha$ instead of Adam; and (*ii*) we use no momentum, $\beta = 0$. We present two results about desirable inverse Hessian learning dynamics and expressiveness, and in Appendix D, some further support from the literature for desirable properties of LODO's inverse Hessian approximation error.

### 4.1 HESSIAN LEARNING DYNAMICS (LODO CAN LEARN THE INVERSE HESSIAN)

In this section, we show that with certain assumptions and approximations, $G(\boldsymbol{\theta}_t)$ converges to the inverse Hessian in a stochastic setting. We restrict our analysis to a version of LODO simplified from Algorithm 1, where the parameterization of $G(\boldsymbol{\theta}_t) \in \mathbb{R}^{n \times n}$ is instead a rearrangement of the parameter vector $\boldsymbol{\theta}_t \in \mathbb{R}^{n^2}$ and is not necessarily symmetric, though the experiment of Section 5.1 supports similar findings for the original version of LODO as well. We also work in the limit of small learning rate $\alpha$. The problem setup is of a noisily moving quadratic bowl:

$$\boldsymbol{s}_t \in \mathbb{R}^n, \quad \boldsymbol{s}_t \overset{iid}{\sim} \mathcal{S}, \quad \mathbb{E}[\boldsymbol{s}_t] = 0, \quad \boldsymbol{x}_t^* = \sum_{\tau=1}^{t} \boldsymbol{s}_\tau, \quad \ell_t(\boldsymbol{x}_t) = \frac{1}{2}(\boldsymbol{x}_t - \boldsymbol{x}_t^*)^T \boldsymbol{H} (\boldsymbol{x}_t - \boldsymbol{x}_t^*) \quad (2)$$

where $\boldsymbol{H}$ is a positive-definite symmetric Hessian matrix and $\mathcal{S}$ is some light tailed distribution of zero mean, for example a multivariate normal distribution $\mathcal{N}(\boldsymbol{s}_t; 0, \boldsymbol{\Sigma})$. Lastly, we assume that $||\boldsymbol{I} - G(\boldsymbol{\theta}_t)\boldsymbol{H}||_2 < 1$ for all $t$, that $G(\boldsymbol{\theta}_t)$ never travels too far from $\boldsymbol{H}^{-1}$.

Under these conditions, the gradient of loss with respect to both $\boldsymbol{x}_t$ and $\boldsymbol{\theta}_t$ can be solved for so LODO turns out to follow the training dynamics

$$\boldsymbol{A}_{t+1} = \boldsymbol{A}_t - \alpha \boldsymbol{H} \boldsymbol{b}_{t+1} \boldsymbol{b}_t^T \boldsymbol{H}^2, \qquad \boldsymbol{b}_{t+1} = \boldsymbol{A}_t \boldsymbol{b}_t - \boldsymbol{s}_t, \quad (3)$$

where we define $\boldsymbol{A}_t = \boldsymbol{I} - G(\boldsymbol{\theta}_t)\boldsymbol{H}$ and $\boldsymbol{b}_t = \boldsymbol{x}_t - \boldsymbol{x}_t^*$ to be the errors in estimates of the inverse Hessian and the minimum, respectively. A full derivation is in Appendix A.1. $\boldsymbol{b}_t$ then exponentially decays to a fixed distribution, and the long term dynamics of $\boldsymbol{A}_t$ should not be greatly affected by the quick compensating movement of $\boldsymbol{b}_t$'s distribution. Thus, to determine the movement of $\boldsymbol{A}_t$, we can approximate the dynamics along a short trajectory by fixing $\boldsymbol{A}_t$ to its value at $t = t_0$ to study the settled distribution of $\boldsymbol{b}_t$.[1] Appendix A.2 gives more rigorous justification for this approximation—

---

[1] We think of this phenomenon as similar to the adiabatic approximation of quantum mechanics. $\boldsymbol{A}_t$, the fixed distribution of $\boldsymbol{b}_t$, and the actual distribution of $\boldsymbol{b}_t$ play the roles of the Hamiltonian, the ground state, and the actual state, respectively.

we keep the $\boldsymbol{A}_{t+1}$ update but rewrite the $\boldsymbol{b}_{t+1}$ update as

$$\boldsymbol{b}_{t+1} = \boldsymbol{A}_{t_0}\boldsymbol{b}_t - \boldsymbol{s}_t. \tag{4}$$

The total contribution of all $\boldsymbol{s}_\tau$ for $t_0 \leq \tau \leq t$ to the movement of $\boldsymbol{A}_t$ in the infinite time horizon is then

$$\alpha \boldsymbol{H}\boldsymbol{A}_{t_0} \left( \sum_{n=0}^{\infty} \boldsymbol{A}_{t_0}{}^n \left( \sum_{\tau=t_0}^{t} \boldsymbol{s}_\tau \boldsymbol{s}_\tau^T \right) (\boldsymbol{A}_{t_0}{}^n)^T \right) \boldsymbol{H}^2 \tag{5}$$

with some extra terms multiplied with pairwise products between $\boldsymbol{s}_i$ and $\boldsymbol{s}_j$ ($i \neq j$), and other terms with $\boldsymbol{b}_{t_0}$. In the long term, the average contribution of each $\boldsymbol{s}_\tau$ to the total then converges to

$$\alpha \boldsymbol{H}\boldsymbol{A}_{t_0} \left( \sum_{n=0}^{\infty} \boldsymbol{A}_{t_0}{}^n \mathbb{E}\left[ \boldsymbol{s}_\tau \boldsymbol{s}_\tau^T \right] (\boldsymbol{A}_{t_0}{}^n)^T \right) \boldsymbol{H}^2 \tag{6}$$

by law of large numbers, where any pairwise products of $\boldsymbol{s}_i$ and $\boldsymbol{s}_j$ for $i \neq j$ have no effect due to independence and the mean of $\boldsymbol{s}_t$ being zero, and the effect of $\boldsymbol{b}_{t_0}$ vanishes due to small $\alpha$. The expected error for this convergence decays to zero like $O((t-t_0)^{-1/2})$ due to the light tailedness of $\boldsymbol{s}_\tau$ (eg. when $\boldsymbol{s}_\tau \overset{\text{iid}}{\sim} \mathcal{N}(\boldsymbol{s}_\tau; 0, \boldsymbol{\Sigma})$) leading to the central limit theorem. Expression 6 is then the step direction for the Euler method for approximating the solution to the differential equation

$$\frac{\mathrm{d}\boldsymbol{A}}{\mathrm{d}t} = -\boldsymbol{H}\boldsymbol{A}\sum_{n=0}^{\infty} \boldsymbol{A}^n \mathbb{E}\left[ \boldsymbol{s}\boldsymbol{s}^T \right] (\boldsymbol{A}^n)^T \boldsymbol{H}^2, \tag{7}$$

which can be shown to cause $\boldsymbol{A}$ to flow towards zero such that $\boldsymbol{G}(\boldsymbol{\theta}_t)$ converges to $\boldsymbol{H}^{-1}$.[2] Therefore, it is reasonable to believe that LODO learns the inverse Hessian over time, given small enough learning rate. The Frobenius norm of the error decays faster when magnitudes/norms of $\boldsymbol{H}$ and $\mathbb{E}\left[ \boldsymbol{s}\boldsymbol{s}^T \right]$ are higher, indicating that both curvature of the Hessian and noisy movement of the quadratic bowl's minimum are good for learning the Hessian for fixed $\alpha$, as long as the approximations mentioned stay accurate. One interpretation of this is that when the noise $\boldsymbol{s}$ is zero in a certain direction, the error $\boldsymbol{b}$ in that direction quickly decays to zero so LODO no longer receives any data along that direction in order to learn that direction's curvature. Furthermore if the Hessian $\boldsymbol{H}$ does not penalize deviation in some direction, then the training signal for that direction vanishes.

We may also produce another analysis for when $\boldsymbol{H}$ has a direction of negative curvature, to show that LODO repels saddle points. Here, we instead choose to parameterize $\boldsymbol{G}(\boldsymbol{\theta}_t)$ as a positive-definite symmetric matrix, and take the limit of small learning rate $\alpha$ such that $\boldsymbol{G}(\boldsymbol{\theta}_t)$ is effectively fixed. Then, $\boldsymbol{G}(\boldsymbol{\theta}_t)\boldsymbol{H}$ has some negative eigenvalue $\lambda$ with eigenvector $\boldsymbol{y}$ (see Appendix B for proof), such that $\boldsymbol{A} = \boldsymbol{I} - \boldsymbol{G}(\boldsymbol{\theta}_t)\boldsymbol{H}$ has an eigenvalue of norm greater than 1, meaning that the error vector $\boldsymbol{b}$ blows up exponentially in the $\boldsymbol{y}$ direction. Since $\boldsymbol{G}(\boldsymbol{\theta}_t)\boldsymbol{H}\boldsymbol{y} = \lambda\boldsymbol{y}$, left multiplying by $\boldsymbol{y}^T \boldsymbol{G}(\boldsymbol{\theta}_t)^{-1}$ further shows that $\boldsymbol{y}^T \boldsymbol{H}\boldsymbol{y} < 0$, implying that the direction of saddle point repulsion is one of negative curvature, which decreases the loss.

## 4.2 Representability (LODO can be Expressive)

In this section, we seek to show that our parameterization of $\boldsymbol{G}(\boldsymbol{\theta})$ is highly expressive; LODO may represent a wider range of inverse Hessians than many other methods. In fact, we show that with

---

[2]$\boldsymbol{A}$ flows towards zero because by substituting the eigendecomposition $\boldsymbol{H} = \boldsymbol{U}\boldsymbol{D}\boldsymbol{U}^T$ where $\boldsymbol{U}\boldsymbol{U}^T = \boldsymbol{I}$, and using $\boldsymbol{B} = \boldsymbol{U}^T \boldsymbol{A}\boldsymbol{U}$, we can show that the norm of $\boldsymbol{B}\boldsymbol{D}^{-1}$ decreases over time,

$$\frac{\mathrm{d}\|\boldsymbol{B}\boldsymbol{D}^{-1}\|_F^2}{\mathrm{d}t} = \frac{\mathrm{d}}{\mathrm{d}t}\mathrm{tr}\left( \boldsymbol{D}^{-2}\boldsymbol{B}^T\boldsymbol{B} \right) \tag{8}$$

$$= -2\mathrm{tr}\left( \boldsymbol{B}^T\boldsymbol{D}\boldsymbol{B}\sum_{n=0}^{\infty} \boldsymbol{B}^n \boldsymbol{U}^T \mathbb{E}\left[ \boldsymbol{s}\boldsymbol{s}^T \right] \boldsymbol{U}(\boldsymbol{B}^n)^T \right) \tag{9}$$

$$= -2\left\| \boldsymbol{D}^{1/2}\boldsymbol{B}\left( \sum_{n=0}^{\infty} \boldsymbol{B}^n \boldsymbol{U}^T \mathbb{E}\left[ \boldsymbol{s}\boldsymbol{s}^T \right] \boldsymbol{U}(\boldsymbol{B}^n)^T \right)^{1/2} \right\|_F^2 \leq 0 \tag{10}$$

where the strict equality is only satisfied for $\boldsymbol{A} = 0$.

an increase in parameter count by only a logarithmic factor, our fixed parameterization of $\boldsymbol{G}(\boldsymbol{\theta})$ can reproduce any possible parameterization $\boldsymbol{G}(\boldsymbol{\theta}) = \tilde{\boldsymbol{F}}^T \tilde{\boldsymbol{F}}$ where vector multiplication with $\tilde{\boldsymbol{F}}$ is computable with some sparse linear neural network. Our parameterization is thus capable of representing inverse Hessians from (Baydin et al., 2017) and (Amid et al., 2022) with only $O(\log \tilde{n})$ times more parameters, and (Moskovitz et al., 2019) with only $O(\log^2 \tilde{n})$ times more parameters. Definition 1 creates a function $\epsilon$ to characterize the mixing rate of random transpositions when trying to shuffle lists, while Theorem 1 uses this $\epsilon$ function to lower bound the probability that the $\tilde{\boldsymbol{G}}(\boldsymbol{\theta})$ network in LODO can represent other linear neural networks.

**Definition 1.** Uniformly sample a sequence of $N\tilde{n}/2$ transpositions of two out of $\tilde{n}$ elements, for integers $\tilde{n}/2 \in \mathbb{N}$ and $N \in \mathbb{N}$, with the condition that every successive block of $\tilde{n}/2$ transpositions commutes internally (transpositions can be rearranged within a block). We replace each transposition with the identity operation with probability $1/2$, and then compose the sequence of transpositions/identities to form a permutation. Then, we define the function $\epsilon(N\tilde{n}/2, \tilde{n})$ such that the expected entropy of this permutation, given the original sequence but not given the locations where identities were placed, is $\log \tilde{n}! - \epsilon(N\tilde{n}/2, \tilde{n})$.

**Theorem 1.** *Uniformly sample permutations $\boldsymbol{P}_i$ and create block-diagonal matrices $\boldsymbol{B}(\boldsymbol{\theta}^{(i)})$ where every block is $2 \times 2$, and whose block contents are listed by the parameters $\boldsymbol{\theta}^{(i)}$. Use these to construct the LODO subnetwork $\tilde{\boldsymbol{G}}(\boldsymbol{\theta})$ as in Equation 1 with some depth $N$ and hidden dimension $\tilde{n}$. Construct any linear neural network $\tilde{\boldsymbol{F}}$ with input dimension, output dimension, and number of weights per layer at most $\tilde{n}$, at most $k$ incoming and at most $k$ outgoing weights for every neuron, depth $d$, and any arrangement of weights. Then, there is a probability of at least*

$$1 - \tilde{n}! N \sqrt{\frac{1}{2} \epsilon \left( \frac{\tilde{n}N}{4d(\lceil \log_2 k \rceil + 1)}, \tilde{n} \right)} \tag{11}$$

*that we can make $\tilde{\boldsymbol{G}}(\boldsymbol{\theta}) = \tilde{\boldsymbol{F}}$ for some $\boldsymbol{\theta}$.*

The proof is in Appendix C.

We believe that random transpositions in the style of Definition 1 are a quick way to shuffle, since the Cayley graph over the symmetric group generated by all transpositions has good expansion properties (Konstantinova & Kravchuk, 2022). In other words, we hypothesize that for large $N$ and $\tilde{n}$, we have $\epsilon(N\tilde{n}/2, \tilde{n}) \approx c_1 \tilde{n} e^{-c_2 N \tilde{n}/2}$ for some positive constants $c_1$ and $c_2$. This would imply that the probability that $\tilde{\boldsymbol{G}}(\boldsymbol{\theta})$ can represent all possible $\tilde{\boldsymbol{F}}$ is at least approximately

$$1 - \tilde{n}! N \sqrt{\frac{c_1 \tilde{n}}{2} \exp \left( \frac{-c_2 \tilde{n} N}{4d(\lceil \log_2 k \rceil + 1)} \right)} \tag{12}$$

which can be made to be above $1 - (n!)^c$ for any constant $c$ by using sufficient depth $N$ which goes like $N \propto d(\log_2 k) \log \tilde{n}$, due to Stirling's approximation. Thus we believe that by only being $O((\log_2 k) \log \tilde{n})$ times deeper than $\tilde{\boldsymbol{F}}$, we can make it very likely that our model $\tilde{\boldsymbol{G}}(\boldsymbol{\theta})$ can represent all possible $\tilde{\boldsymbol{F}}$.

## 5 EXPERIMENTS WITH LODO

We present here a number of tasks which provide experimental evidence for the theoretical results we claim, though we test a variety of optimizers on these tasks. Appendix G presents an additional experiment on the Rosenbrock function. Appendix E explains how we tuned each optimizer's hyperparameters for each task. Optimizers were used 8 times for every experiment to ensure reproducibility of results, unless otherwise stated.

### 5.1 NOISY QUADRATIC BOWL

We use various optimizers to track the minimum of a quadratic bowl of fixed true Hessian $\boldsymbol{H}$ as its minimum is perturbed by noise at every step, to demonstrate that LODO correctly learns its inverse Hessian representation as claimed in Section 4.1. The setup of the noisy quadratic bowl is the same

as in Section 4.1 and details are provided in Appendix F.1. We interpret an optimizer to be better if it can maintain a lower loss in the infinite step limit, since the error builds at a constant rate over time and the optimizer's role is to react and correct it as quickly as possible. We tested each optimizer by using it to track the minimum of the moving quadratic bowl over 100k steps. Learning curves in Figure 2 and losses in Table 5 of Appendix F.1 show that LODO tracks the minimum more accurately than all other optimizers, and that an estimator of the inverse Hessian approximation error $||\boldsymbol{I} - \boldsymbol{G}(\boldsymbol{\theta}_t)\boldsymbol{H}||_F^2/n$ decays over time. Other optimizers underperform because their preconditioners are less expressive, being either diagonal or low-rank and thus unable to capture the off-diagonal elements or non-top-few singular values of the true inverse Hessian, leading to a higher loss.

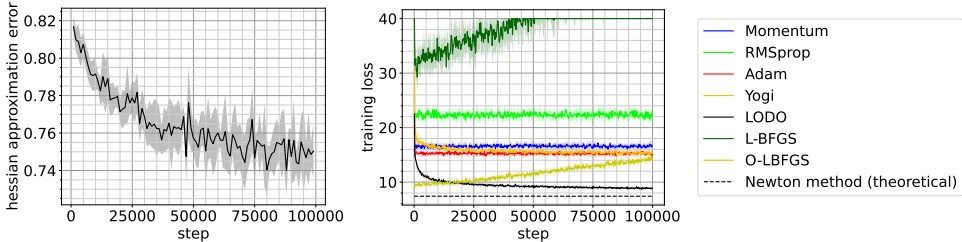

Figure 2: **Left:** LODO's average inverse Hessian approximation error $\sigma = \sqrt{||\boldsymbol{I} - \boldsymbol{G}(\boldsymbol{\theta}_t)\boldsymbol{H}||_F^2/n}$ on the noisy quadratic bowl task of Section 5.1. $\sigma^2$ is measured by the unbiased estimator $\frac{1}{100}\sum_{i=1}^{100} ||(\boldsymbol{I} - \boldsymbol{G}(\boldsymbol{\theta}_t)\boldsymbol{H})\boldsymbol{v}_i||_2^2$ with random independent unit vectors $\boldsymbol{v}_i$. **Right:** Average training loss learning curves, smoothed by averaging over blocks of 200 steps each. The dotted line shows the theoretically best possible loss using Newton's method. A better optimizer maintains a lower loss after infinite steps, since for this task, loss is introduced over time and the optimizer serves to quickly suppress it. **Left & Right:** Error margins indicate $\pm 1$ standard deviation between the performances of 8 optimizers at every step. The version of L-BFGS is one with stochastic modifications from (Schraudolph et al., 2007), instead of the original from (Nocedal & Wright, 1999).

## 5.2 IMAGE GENERATION

We design a challenge for LODO—an autoregressive image generator for MNIST. We intend to demonstrate the effectiveness of LODO at scale when used to train a semi-realistic neural network. The task is similar to training a PixelCNN (Oord et al., 2016) to generate MNIST images (Lecun et al., 1998), and is fully described in Appendix F.2. We tested LODO alongside a variety optimizers for 300k steps, though we could not get any quasi-Newton methods to converge on this task. Learning curves in Figures 3 and losses in Table 1 show that LODO trains at a speed that is competitive against other optimizers. Figure 7 in Appendix F.2 shows some MNIST digits we generated.

Table 1: Negative log likelihoods in nats per pixel after training for 300k steps on the MNIST image generation task of Section 5.2 with every optimizer. Values are averaged over the last 10% of training before the stated training milestone, with means and standard deviations produced across 8 optimization runs with different randomization seeds. The top 3 optimizers are underlined for each metric. Our timing setup is described in Appendix H.

| | Training loss | | Test loss | | |
| --- | --- | --- | --- | --- | --- |
| Optimizer | 300k steps | 50k sec. ($\sim$ 14 h.) | 300k steps | 50k sec. | Steps / sec. |
| Adam | $0.830 \pm 0.005$ | $0.859 \pm 0.009$ | 0.809 | 0.854 | $7.08 \pm 0.03$ |
| Momentum | $0.708 \pm 0.005$ | $\underline{0.698 \pm 0.005}$ | $\underline{0.689}$ | $\underline{0.685}$ | $7.10 \pm 0.03$ |
| RMSprop | $0.917 \pm 0.010$ | $0.920 \pm 0.014$ | 0.931 | 0.899 | $7.10 \pm 0.02$ |
| Yogi | $\underline{0.683 \pm 0.002}$ | $\underline{0.677 \pm 0.003}$ | $\underline{0.686}$ | $\underline{0.674}$ | $7.42 \pm 0.02$ |
| LARS (You et al., 2017) | $\underline{0.701 \pm 0.006}$ | $0.702 \pm 0.006$ | $\underline{0.688}$ | $\underline{0.688}$ | $5.89 \pm 0.02$ |
| LODO (ours) | $\underline{0.696 \pm 0.004}$ | $\underline{0.698 \pm 0.005}$ | $\underline{0.689}$ | 0.689 | $5.64 \pm 0.02$ |

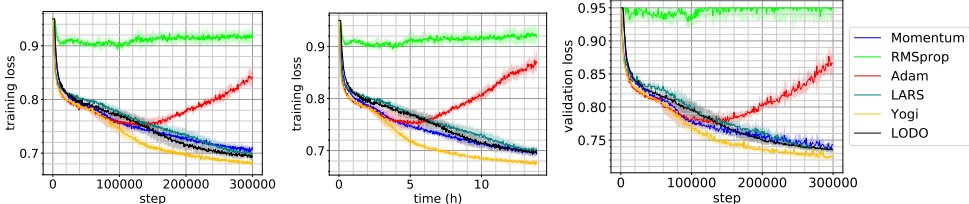

Figure 3: Average training loss learning curves over 8 training runs on the MNIST image generation task of Section 5.2. Learning curves are smoothed by averaging over blocks of 600 steps each. Error margins indicate $\pm 1$ standard deviation between the performances of the 8 optimizers at every step. **Left:** By step. **Middle:** By time. Our timing setup is described in Appendix H. **Right:** Validataion loss by step, using a subset of 64 images excluded from the training data. Each image provides 784 pixel colors to predict, so the validation dataset effectively consists of 50176 samples.

Table 2: Negative log likelihoods in nats per pixel after training for 300k steps on the MNIST image generation task of Section 5.2 with ablated versions of LODO from Section 5.2. Values are averaged over the last 10% of training, with means and standard deviations produced across 8 runs with different randomization seeds. Our timing setup is described in Appendix H. *Setup is similar to (Amid et al., 2022) **(Baydin et al., 2017).

| | Training loss | | Test loss | | |
| Optimizer | 300k steps | 50k sec. ($\sim 14$ h.) | 300k steps | 50k sec. | Steps / sec. |
|---|---|---|---|---|---|
| LODO | $0.696 \pm 0.004$ | $0.698 \pm 0.005$ | 0.689 | 0.689 | $5.64 \pm 0.02$ |
| LODO-Diagonal* | Diverged | Diverged | Diverged | Diverged | $9.92 \pm 0.09$ |
| LODO-Global** | $0.770 \pm 0.035$ | $0.919 \pm 0.139$ | 0.747 | 0.801 | $9.92 \pm 0.03$ |
| LODO-Residuals | $0.701 \pm 0.004$ | $0.750 \pm 0.008$ | 0.693 | 0.741 | $3.31 \pm 0.03$ |
| LODO-No-Momentum | $0.753 \pm 0.007$ | $0.756 \pm 0.007$ | 0.750 | 0.752 | $5.46 \pm 0.06$ |
| LODO-SGD | $0.709 \pm 0.004$ | $0.714 \pm 0.004$ | 0.698 | 0.707 | $5.44 \pm 0.02$ |

We study the contributions of various components of LODO to its performance by replacing components to observe a performance change, as detailed below. For example, we replace the Adam meta-optimizer with SGD to form a new version of LODO (which we call "LODO-SGD"); its performance is shown in Table 2. Other modifications are listed below.

### 5.2.1 EIGENDECOMPOSED ALTERNATE PARAMETERIZATION

We would like to show that architecture LODO uses to choose the step is flexible. We modify the matrix decomposition in Section 3 to $\boldsymbol{G}(\boldsymbol{\theta}) = \alpha_0 \left(\boldsymbol{I} \quad 0\right) \tilde{\boldsymbol{G}}(\boldsymbol{\theta})^T \boldsymbol{D} \tilde{\boldsymbol{G}}(\boldsymbol{\theta}) \left(\boldsymbol{I} \quad 0\right)^T$ by adding learned diagonal matrix $\boldsymbol{D}$ in the middle and changing $\tilde{\boldsymbol{G}}(\boldsymbol{\theta})$ to a product of many weighted permutations each added to the identity matrix. The neural network which $\tilde{\boldsymbol{G}}(\boldsymbol{\theta})$ represents now has residual connections, and the initialization is modified accordingly. Losses in Table 2 show that this version (which we call "LODO-Residuals") performs only slightly worse than LODO for the same number of steps, reflecting the flexibility in the design of LODO.

### 5.2.2 SIMPLER APPROXIMATE HESSIANS

We presumed that the representability result of Section 4.2 is only useful because LODO's strength comes from the flexibility that $\boldsymbol{G}(\boldsymbol{\theta})$ gives in configuring pairwise interactions between parameters. We therefore expect that using a simpler form of Hessian should hurt the performance of LODO. We test two simpler forms of $\boldsymbol{G}(\boldsymbol{\theta})$: $\boldsymbol{G}(\boldsymbol{\theta}) = \alpha_0 \text{diag}(\boldsymbol{\theta})$ (which we call "LODO-Diagonal") for $\boldsymbol{\theta} \in \mathbb{R}^n$ initialized to a vector of ones, similar to (Amid et al., 2022)—and the even simpler $\boldsymbol{G}(\boldsymbol{\theta}) = \alpha_0 \theta \boldsymbol{I}$ (which we call "LODO-Global") for $\theta \in \mathbb{R}$ initialized to 1, as in (Baydin et al., 2017). Losses in Table 2 show that the original version of LODO performs the best, verifying our hypothesis.

### 5.2.3 Effects of Using EMAs of Gradients

Similarly to how momentum works for SGD, LODO's input gradients are preproccessed by accumulation into EMAs. To test our claim in Section 3 that momentum benefits LODO, we try 8 separate momentum decay rates in a logarithmic grid from no momentum to the optimal amount of momentum found ($\beta = 0.9343$), and test each decay rate once. Figure 4 shows a clear trend that at least up to the optimal decay rate, increasing the effect of momentum improves LODO. We also try removing momentum completely (we call the modified version "LODO-No-Momentum"); results are shown in Table 2.

## 6 Discussion

LODO is a middle point between L2O methods and quasi-Newton methods, retaining significant advantages of both classes of optimizers. Via LODO, we bring ideas from both classes of optimization methods to offer potential solutions to problems common on the other side.

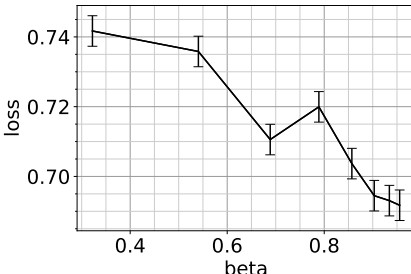

Figure 4: LODO's training loss as a function of the momentum decay coefficient $\beta$, averaged over the last 10% of 300k steps, for the image generation task of Section 5.2. Momentum improves LODO. Error bars depict LODO's uncertainty from Table 1.

Relative to quasi-Newton methods, LODO offers advantages associated with the use of a meta-optimizer on a neural optimizer. Crucially, LODO determines its inverse Hessian estimate using all past gradients, whereas most other quasi-Newton methods use a finite history of them. This allows LODO to retain information about the inverse Hessian for much longer than other methods. This is useful if the gradients contain enough noise that useful signals can only be obtained by accumulating information from many gradient samples. Our theory further shows that the linear neural network in LODO is optimal to a certain extent: it can probably represent all linear neural networks smaller by a logarithmic factor—allowing a huge class of inverse Hessians. Our image generation task demonstrates that LODO succeeds in a semi-realistic stochastic nonconvex task where other quasi-Newton optimizers diverge. Due to our use of L2O, LODO also has flexibility in the design of its linear neural network, which makes it amenable to further research and refinement.

Relative to L2O, LODO offers advantages associated with the restructuring of the outer and inner loop into a single loop. Most importantly, our modification to L2O alleviates the requirement for meta-training time and the training task distribution. This is at the cost of increased inner loop unrolling truncation bias, but it takes advantage of this sacrifice in resolving the need to compute second-order gradients. LODO still inherits issues of high memory usage and slow step computation from L2O methodology though. Our theory offers some understanding of how LODO learns to optimize, which is rare for L2O methods: the Hessian approximation error decays as learning progresses. We import the idea from quasi-Newton methods that the gradient of one parameter can affect the step for another, which comes from the presence of off-diagonal elements in the Hessian. As shown in Section 4.2, LODO presents an efficient way of approximating subsets of the $O(n^2)$ possible pairwise parameter interactions in $O(n \log n)$ time. Such interactions are commonly ignored in the design of L2O and more mainstream optimizers, yet our image generation task demonstrates their importance, as evidenced by the improved performance of LODO over ablated versions as well as SGD.

**Conclusion** Through LODO, we provide a new way of using L2O methods online without any meta-training to perform quasi-Newton optimization. We introduce the strengths and advantages of quasi-Newton methods and L2O to each other and combine them in a harmonious manner. LODO's abilities showcase the applicability of online L2O methods with nested optimizers to the training of modern neural networks. Our unique methodology serves as a stepping stone for the further development and use of L2O in quasi-Newton optimization and vice versa.

## 7 REPRODUCIBILITY OF RESULTS

We have described all the details of the construction of our optimizer in Section 3, and all its modified versions in Section 5.2. All the tasks used for experiments are introduced in Section 5 and are fully described in Appendices F.1 and F.2. To further ensure reproducibility, we have also listed all the hyperparameters found for each optimizer and task in Appendix E. All our training runs were repeated 8 times to verify that each measurement of empirical behavior is consistent, with low standard deviation. Furthermore, we will release the code for all our optimizers and experiments on Github for others to see.

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

# A  ELABORATION ON HESSIAN LEARNING DYNAMICS

## A.1  DERIVATION OF TRAINING DYNAMICS

This section gives a derivation of the result that under the problem setup of Section 4.1, LODO follows the Hessian learning dynamics

$$\boldsymbol{A}_{t+1} = \boldsymbol{A}_t - \alpha \boldsymbol{H} \boldsymbol{b}_{t+1} \boldsymbol{b}_t^T \boldsymbol{H}^2, \qquad \boldsymbol{b}_{t+1} = \boldsymbol{A}_t \boldsymbol{b}_t - \boldsymbol{s}_t, \tag{3}$$

where $\boldsymbol{A}_t = \boldsymbol{I} - \boldsymbol{G}(\boldsymbol{\theta}_t)\boldsymbol{H}$ and $\boldsymbol{b}_t = \boldsymbol{x}_t - \boldsymbol{x}_t^*$ as long as $\boldsymbol{G}(\boldsymbol{\theta}_t)$ is parameterized as a dense matrix filled with elements of $\boldsymbol{\theta}_t$, and no momentum is used.

We first let $\boldsymbol{b}_t$ be $\boldsymbol{x}_t - \boldsymbol{x}_t^*$. The loss at time $t$ is then

$$\ell_t = \frac{1}{2} \boldsymbol{b}_t^T \boldsymbol{H} \boldsymbol{b}_t. \tag{13}$$

The gradient is then computed to be

$$\frac{\mathrm{d}\ell}{\mathrm{d}\boldsymbol{x}_t} = \boldsymbol{H} \boldsymbol{b}_t. \tag{14}$$

The step taken then produces the next parameters:

$$\boldsymbol{x}_{t+1} = \boldsymbol{x}_t - \boldsymbol{G}(\boldsymbol{\theta}_t)\boldsymbol{H}\boldsymbol{b}_t. \tag{15}$$

Subtracting $\boldsymbol{x}_{t+1}^* = \boldsymbol{x}_t^* + \boldsymbol{s}_t$, we get the recurrence for $\boldsymbol{b}_t$,

$$\boldsymbol{x}_{t+1} - \boldsymbol{x}_{t+1}^* = \boldsymbol{x}_t - \boldsymbol{x}_t^* - \boldsymbol{s}_t - \boldsymbol{G}(\boldsymbol{\theta}_t)\boldsymbol{H}\boldsymbol{b}_t \tag{16}$$

$$\boldsymbol{b}_{t+1} = \boldsymbol{b}_t - \boldsymbol{G}(\boldsymbol{\theta}_t)\boldsymbol{H}\boldsymbol{b}_t - \boldsymbol{s}_t \tag{17}$$

$$= (\boldsymbol{I} - \boldsymbol{G}(\boldsymbol{\theta}_t)\boldsymbol{H})\boldsymbol{b}_t - \boldsymbol{s}_t \tag{18}$$

$$= \boldsymbol{A}_t \boldsymbol{b}_t - \boldsymbol{s}_t. \tag{3}$$

The loss at time $t + 1$ is computed to be

$$\ell_{t+1} = \frac{1}{2} \boldsymbol{b}_{t+1}^T \boldsymbol{H} \boldsymbol{b}_{t+1} \tag{19}$$

$$= \frac{1}{2} (\boldsymbol{A}_t \boldsymbol{b}_t - \boldsymbol{s}_t)^T \boldsymbol{H} (\boldsymbol{A}_t \boldsymbol{b}_t - \boldsymbol{s}_t) \tag{20}$$

$$= \frac{1}{2} ((\boldsymbol{I} - \boldsymbol{G}(\boldsymbol{\theta}_t)\boldsymbol{H})\boldsymbol{b}_t - \boldsymbol{s}_t)^T \boldsymbol{H} ((\boldsymbol{I} - \boldsymbol{G}(\boldsymbol{\theta}_t)\boldsymbol{H})\boldsymbol{b}_t - \boldsymbol{s}_t). \tag{21}$$

LODO also computes a step of $\boldsymbol{\theta}_t$ using the loss on the next step. Since the elements of $\boldsymbol{\theta}_t$ are just a rearrangement of the elements of $\boldsymbol{G}(\boldsymbol{\theta}_t)$ in our derivation, an update of $\boldsymbol{\theta}_t$ can be treated instead like an update of $\boldsymbol{G}(\boldsymbol{\theta}_t)$. The gradient of $\ell_{t+1}$ with respect to $\boldsymbol{G}(\boldsymbol{\theta}_t)$ is then computed to be

$$\frac{\mathrm{d}\ell_{t+1}}{\mathrm{d}\boldsymbol{G}(\boldsymbol{\theta}_t)} = -\boldsymbol{H}((\boldsymbol{I} - \boldsymbol{G}(\boldsymbol{\theta}_t)\boldsymbol{H})\boldsymbol{b}_t - \boldsymbol{s}_t)\boldsymbol{b}_t^T \boldsymbol{H} \tag{22}$$

$$= -\boldsymbol{H}(\boldsymbol{A}_t \boldsymbol{b}_t - \boldsymbol{s}_t)\boldsymbol{b}_t^T \boldsymbol{H} \tag{23}$$

$$= -\boldsymbol{H}\boldsymbol{b}_{t+1}\boldsymbol{b}_t^T \boldsymbol{H} \tag{24}$$

and the step of $\boldsymbol{G}(\boldsymbol{\theta}_t)$ is

$$\boldsymbol{G}(\boldsymbol{\theta}_{t+1}) = \boldsymbol{G}(\boldsymbol{\theta}_t) + \alpha \boldsymbol{H} \boldsymbol{b}_{t+1} \boldsymbol{b}_t^T \boldsymbol{H} \tag{25}$$

resulting in the recurrence for $\boldsymbol{A}_t$:

$$\boldsymbol{A}_{t+1} = \boldsymbol{I} - \boldsymbol{G}(\boldsymbol{\theta}_{t+1})\boldsymbol{H} \tag{26}$$

$$= \boldsymbol{I} - (\boldsymbol{G}(\boldsymbol{\theta}_t) + \alpha \boldsymbol{H} \boldsymbol{b}_{t+1} \boldsymbol{b}_t^T \boldsymbol{H})\boldsymbol{H} \tag{27}$$

$$= \boldsymbol{A}_t - \alpha \boldsymbol{H} \boldsymbol{b}_{t+1} \boldsymbol{b}_t^T \boldsymbol{H}^2. \tag{3}$$

### A.2 VALIDITY OF APPROXIMATION ARGUMENT

This section gives justification for the approximation in Section 4.1 of the long term trajectory of the recurrence

$$\boldsymbol{A}_{t+1} = \boldsymbol{A}_t - \alpha \boldsymbol{H} \boldsymbol{b}_{t+1} \boldsymbol{b}_t^T \boldsymbol{H}^2 \tag{28}$$

$$\boldsymbol{b}_{t+1} = \boldsymbol{A}_t \boldsymbol{b}_t - \boldsymbol{s}_t \tag{29}$$

by replacing with

$$\boldsymbol{A}'_{t+1} = \boldsymbol{A}'_t - \alpha \boldsymbol{H} \boldsymbol{b}'_{t+1} \boldsymbol{b}'^T_t \boldsymbol{H}^2 \tag{30}$$

$$\boldsymbol{b}'_{t+1} = \boldsymbol{A}'_{t_0} \boldsymbol{b}'_t - \boldsymbol{s}_t \tag{31}$$

when $\alpha$ is small and the initial conditions at $t_0$ are the same: $\boldsymbol{A}_{t_0} = \boldsymbol{A}'_{t_0}$ and $\boldsymbol{b}_{t_0} = \boldsymbol{b}'_{t_0} = 0$. We will work in the bounded noise case $||\boldsymbol{s}_t||_2 < \infty$, where $||\boldsymbol{b}'_t||_2$ is upper bounded by some $||\boldsymbol{A}'_{t_0}||_2$ dependent constant $b_{\max}$ due to exponential decay in Equation 31. In the case where the noise is not bounded, a probabilistic analysis can be done instead, though we do not provide one.

To justify this approximation, we prove that the spectral norm of long term deviation corrected for learning rate is small over short distances $r$, in the following theorem:

**Theorem 2.**

$$\lim_{r \to 0} \lim_{\alpha \to 0} \frac{1}{r} ||\boldsymbol{A}_{t_0 + \lfloor r/\alpha \rfloor} - \boldsymbol{A}'_{t_0 + \lfloor r/\alpha \rfloor}||_2 = 0. \tag{32}$$

In other words, the local movement of $\boldsymbol{A}$ rescaled for learning rate is unaffected by our approximation when the learning rate $\alpha$ is small.

*Proof.* Our proof strategy is as follows:

1. We will first define variables to denote bounds on the movement of $\boldsymbol{A}$ and the approximation error in $\boldsymbol{b}$.

2. We will show that these variables bound each other, and then we will combine these bounds to create a single recursive bound on the movement of $\boldsymbol{A}$.

3. We will characterize the bound's growth and it will turn out that $\boldsymbol{A}$ has a maximum movement speed along any trajectory of sufficiently short length.

4. Due to the slow movement of $\boldsymbol{A}$, we can deduce that the approximation error in $\boldsymbol{b}$ increases at a bounded rate.

5. Since approximation errors in $\boldsymbol{A}$ are an accumulation of errors in $\boldsymbol{b}$, we will show that deviation between the true and approximate $\boldsymbol{A}$ trajectories is quadratic in the distance along the trajectory.

6. We conclude that the approximation error vanishes for short trajectories and small learning rates.

**First part.** We first define the maximum drift in $\boldsymbol{A}$

$$\epsilon_{\boldsymbol{A}, t_0 + \Delta t} = \max_{t_0 \leq \tau \leq t_0 + \Delta t} ||\boldsymbol{A}_\tau - \boldsymbol{A}'_{t_0}||_2 \tag{33}$$

up to time difference $\Delta t$ for $0 \leq \Delta t \leq R/\alpha$ for some chosen small constant trajectory length $R > 0$. We will pick $R$ later. We will also define the maximum error in $\boldsymbol{b}$

$$\epsilon_{\boldsymbol{b}, t_0 + \Delta t} = \max_{t_0 \leq \tau \leq t_0 + \Delta t} ||\boldsymbol{b}_\tau - \boldsymbol{b}'_\tau||_2 \tag{34}$$

up to the same time.

**Second part.** For the bound in one direction, we have that for all $\tau$ such that $t_0 \leq \tau \leq t_0 + \Delta t$,

$$||\boldsymbol{b}_{\tau+1} - \boldsymbol{b}'_{\tau+1}||_2 = ||\boldsymbol{A}_\tau \boldsymbol{b}_\tau - \boldsymbol{s}_\tau - (\boldsymbol{A}'_{t_0} \boldsymbol{b}'_\tau - \boldsymbol{s}_\tau)||_2 \tag{35}$$

$$= ||\boldsymbol{A}_\tau \boldsymbol{b}_\tau - \boldsymbol{A}'_{t_0} \boldsymbol{b}'_\tau||_2 \tag{36}$$

$$\leq ||\boldsymbol{A}_\tau \boldsymbol{b}_\tau - \boldsymbol{A}'_{t_0} \boldsymbol{b}_\tau||_2 + ||\boldsymbol{A}'_{t_0} \boldsymbol{b}_\tau - \boldsymbol{A}'_{t_0} \boldsymbol{b}'_\tau||_2 \tag{37}$$

$$\leq ||\boldsymbol{A}_\tau - \boldsymbol{A}'_{t_0}||_2 ||\boldsymbol{b}_\tau||_2 + ||\boldsymbol{A}'_{t_0}||_2 ||\boldsymbol{b}_\tau - \boldsymbol{b}'_\tau||_2 \tag{38}$$

$$\leq \epsilon_{\boldsymbol{A}, t_0 + \Delta t} ||\boldsymbol{b}_\tau||_2 + ||\boldsymbol{A}'_{t_0}||_2 ||\boldsymbol{b}_\tau - \boldsymbol{b}'_\tau||_2 \tag{39}$$

using the triangle inequality and sub-multiplicativity for the spectral norm $||\cdot||_2$. This is a recurrence in $||\boldsymbol{b}_\tau - \boldsymbol{b}'_\tau||_2$; by induction we have that for $t_0 \leq \tau \leq t_0 + \Delta t + 1$,

$$||\boldsymbol{b}_\tau - \boldsymbol{b}'_\tau||_2 \leq \epsilon_{\boldsymbol{A}, t_0 + \Delta t} \sum_{\tau_1 = t_0}^{\tau - 1} ||\boldsymbol{A}'_{t_0}||_2^{\tau - 1 - \tau_1} ||\boldsymbol{b}_{\tau_1}||_2 \tag{40}$$

such that we produce the bound

$$\epsilon_{\boldsymbol{b}, t_0 + \Delta t + 1} \leq \epsilon_{\boldsymbol{A}, t_0 + \Delta t} \max_{t_0 \leq \tau \leq t_0 + \Delta t + 1} \sum_{\tau_1 = t_0}^{\tau - 1} ||\boldsymbol{A}'_{t_0}||_2^{\tau - 1 - \tau_1} ||\boldsymbol{b}_{\tau_1}||_2 \tag{41}$$

$$\leq \epsilon_{\boldsymbol{A}, t_0 + \Delta t} \max_{t_0 \leq \tau \leq t_0 + \Delta t + 1} \sum_{\tau_1 = t_0}^{\tau - 1} ||\boldsymbol{A}'_{t_0}||_2^{\tau - 1 - \tau_1} (||\boldsymbol{b}'_{\tau_1}||_2 + \epsilon_{\boldsymbol{b}, t_0 + \Delta t}) \tag{42}$$

$$\leq \epsilon_{\boldsymbol{A}, t_0 + \Delta t} \sum_{\tau_1 = t_0}^{t_0 + \Delta t} ||\boldsymbol{A}'_{t_0}||_2^{t_0 + \Delta t - \tau_1} (b_{\max} + \epsilon_{\boldsymbol{b}, t_0 + \Delta t}) \tag{43}$$

$$\leq \epsilon_{\boldsymbol{A}, t_0 + \Delta t} \frac{\epsilon_{\boldsymbol{b}, t_0 + \Delta t + 1} + b_{\max}}{1 - ||\boldsymbol{A}'_{t_0}||_2} \tag{44}$$

$$\epsilon_{\boldsymbol{b}, t_0 + \Delta t + 1} \leq \frac{\epsilon_{\boldsymbol{A}, t_0 + \Delta t} b_{\max}}{1 - ||\boldsymbol{A}'_{t_0}||_2 - \epsilon_{\boldsymbol{A}, t_0 + \Delta t}}. \tag{45}$$

Now, we show a reverse bound: for all $\tau$ such that $t_0 \leq \tau \leq t_0 + \Delta t$, we have

$$||\boldsymbol{A}_{\tau+1} - \boldsymbol{A}'_{t_0}||_2 = ||\boldsymbol{A}_\tau - \alpha \boldsymbol{H} \boldsymbol{b}_{\tau+1} \boldsymbol{b}_\tau^T \boldsymbol{H}^2 - \boldsymbol{A}'_{t_0}||_2 \tag{46}$$

$$\leq ||\boldsymbol{A}_\tau - \boldsymbol{A}'_{t_0}||_2 + \alpha ||\boldsymbol{H}||_2^3 ||\boldsymbol{b}_\tau||_2 ||\boldsymbol{b}_{\tau+1}||_2 \tag{47}$$

$$\leq ||\boldsymbol{A}_\tau - \boldsymbol{A}'_{t_0}||_2 + \alpha ||\boldsymbol{H}||_2^3 (||\boldsymbol{b}'_\tau||_2 + \epsilon_{\boldsymbol{b}, t_0 + \Delta t + 1})(||\boldsymbol{b}'_{\tau+1}||_2 + \epsilon_{\boldsymbol{b}, t_0 + \Delta t + 1}) \tag{48}$$

$$\leq ||\boldsymbol{A}_\tau - \boldsymbol{A}'_{t_0}||_2 + \alpha ||\boldsymbol{H}||_2^3 (b_{\max} + \epsilon_{\boldsymbol{b}, t_0 + \Delta t + 1})^2 \tag{49}$$

By induction we have for $t_0 \leq \tau \leq t_0 + \Delta t + 1$,

$$||\boldsymbol{A}_\tau - \boldsymbol{A}'_{t_0}||_2 \leq \alpha ||\boldsymbol{H}||_2^3 (\tau - t_0)(b_{\max} + \epsilon_{\boldsymbol{b}, t_0 + \Delta t + 1})^2 \tag{50}$$

such that we produce the reverse bound

$$\epsilon_{\boldsymbol{A}, t_0 + \Delta t + 1} \leq \alpha ||\boldsymbol{H}||_2^3 (\Delta t + 1)(b_{\max} + \epsilon_{\boldsymbol{b}, t_0 + \Delta t + 1})^2. \tag{51}$$

**Third part.** Substituting the bound in Equation 45 into the bound in Equation 51, we produce the recurrence

$$\epsilon_{\boldsymbol{A}, t_0 + \Delta t + 1} \leq \alpha ||\boldsymbol{H}||_2^3 b_{\max}^2 (\Delta t + 1) \left(1 + \frac{\epsilon_{\boldsymbol{A}, t_0 + \Delta t}}{1 - ||\boldsymbol{A}'_{t_0}||_2 - \epsilon_{\boldsymbol{A}, t_0 + \Delta t}}\right)^2 \tag{52}$$

$$= \alpha ||\boldsymbol{H}||_2^3 b_{\max}^2 (\Delta t + 1) \left(\frac{1 - ||\boldsymbol{A}'_{t_0}||_2}{1 - ||\boldsymbol{A}'_{t_0}||_2 - \epsilon_{\boldsymbol{A}, t_0 + \Delta t}}\right)^2 \tag{53}$$

$$= f(\epsilon_{\boldsymbol{A}, t_0 + \Delta t}). \tag{54}$$

where

$$f(x) = \alpha ||\boldsymbol{H}||_2^3 b_{\max}^2 (\Delta t + 1) \left(\frac{1 - ||\boldsymbol{A}'_{t_0}||_2}{1 - ||\boldsymbol{A}'_{t_0}||_2 - x}\right)^2. \tag{55}$$

To bound the movement of $\boldsymbol{A}$, we must use the fact that when

$$0 \leq \Delta t \leq \frac{4}{27} \frac{1 - ||\boldsymbol{A}'_{t_0}||_2}{\alpha ||\boldsymbol{H}||_2^3 b_{\max}^2} - 1 \tag{56}$$

the function $f$ maps the interval

$$I_{\Delta t} = \left[0, \frac{9}{4}\alpha ||\boldsymbol{H}||_2^3 b_{\max}^2 (\Delta t + 1)\right] \subseteq \left[0, \frac{1}{3}(1 - ||\boldsymbol{A}'_{t_0}||_2)\right] \tag{57}$$

to a subset of itself. Since at $\Delta t = 0$ we have $\epsilon_{\boldsymbol{A}, t_0 + \Delta t} = 0 \in I_{\Delta t}$, and we also have $I_{\Delta t} \subseteq I_{\Delta t + 1}$, we may deduce by induction on $\Delta t$ that $\epsilon_{\boldsymbol{A}, t_0 + \Delta t} \in I_{\Delta t}$ as long as Equation 56 holds, and thus there is a bound

$$\epsilon_{\boldsymbol{A}, t_0 + \Delta t} \leq \frac{9}{4}\alpha ||\boldsymbol{H}||_2^3 b_{\max}^2 (\Delta t + 1) \leq \frac{1}{3}(1 - ||\boldsymbol{A}'_{t_0}||_2) \tag{58}$$

on the movement speed of $\boldsymbol{A}$ as long as Equation 56 holds.

**Fourth part.** Note that we have assumed that $0 \leq \Delta t \leq R/\alpha$ for some constant $R$ which we have not yet picked. By choosing

$$R \leq \frac{4}{27} \frac{1 - ||\boldsymbol{A}'_{t_0}||_2}{||\boldsymbol{H}||_2^3 b_{\max}^2} - \alpha \tag{59}$$

we may always guarantee Equation 56, which implies Equation 58. Then when Equation 58 is substituted into Equation 45, we create a small bound on the approximation error in $\boldsymbol{b}$ which begins at zero and increases with time,

$$\epsilon_{\boldsymbol{b}, t_0 + \Delta t + 1} \leq \frac{\frac{9}{4}\alpha ||\boldsymbol{H}||_2^3 b_{\max}^3 (\Delta t + 1)}{1 - ||\boldsymbol{A}'_{t_0}||_2 - \frac{9}{4}\alpha ||\boldsymbol{H}||_2^3 b_{\max}^2 (\Delta t + 1)} \tag{60}$$

$$\leq \frac{\alpha b_{\max}(\Delta t + 1)}{3R - \alpha(\Delta t + 1)} \tag{61}$$

for $0 \leq \Delta t \leq R/\alpha$. This also holds trivially for $\Delta t = -1 \implies \epsilon_{\boldsymbol{b}, t_0 + \Delta t + 1} = 0$, so we may re-index to have

$$\epsilon_{\boldsymbol{b}, t_0 + \Delta t} \leq \frac{\alpha b_{\max} \Delta t}{3R - \alpha \Delta t} \tag{62}$$

for $0 \leq \Delta t \leq R/\alpha + 1$. Since the right side of Equation 62 is convex in $\Delta t$ over $\Delta t \in [0, R/\alpha]$, we may bound by a linear function with the same endpoints

$$\epsilon_{\boldsymbol{b}, t_0 + \Delta t} \leq \frac{\alpha b_{\max}}{2R} \Delta t \tag{63}$$

for $0 \leq \Delta t \leq R/\alpha$.

**Fifth part.** Finally, we use this bound on approximation error in $\boldsymbol{b}$ to bound approximation error in $\boldsymbol{A}$.

$$||\boldsymbol{A}_{t_0 + \Delta t + 1} - \boldsymbol{A}'_{t_0 + \Delta t + 1}||_2$$

$$= ||\boldsymbol{A}_{t_0 + \Delta t} - \alpha \boldsymbol{H} \boldsymbol{b}_{t_0 + \Delta t + 1} \boldsymbol{b}_{t_0 + \Delta t}^T \boldsymbol{H}^2 - (\boldsymbol{A}'_{t_0 + \Delta t} - \alpha \boldsymbol{H} \boldsymbol{b}'_{t_0 + \Delta t + 1} \boldsymbol{b}'^T_{t_0 + \Delta t} \boldsymbol{H}^2)||_2 \tag{64}$$

$$\leq ||\boldsymbol{A}_{t_0 + \Delta t} - \boldsymbol{A}'_{t_0 + \Delta t}||_2 + \alpha ||\boldsymbol{H}||^3 ||\boldsymbol{b}_{t_0 + \Delta t + 1} \boldsymbol{b}_{t_0 + \Delta t}^T - \boldsymbol{b}'_{t_0 + \Delta t + 1} \boldsymbol{b}'^T_{t_0 + \Delta t}||_2 \tag{65}$$

$$\leq ||\boldsymbol{A}_{t_0 + \Delta t} - \boldsymbol{A}'_{t_0 + \Delta t}||_2 + \alpha ||\boldsymbol{H}||^3 \bigg( ||\boldsymbol{b}_{t_0 + \Delta t + 1}||_2 ||\boldsymbol{b}_{t_0 + \Delta t}^T - \boldsymbol{b}'^T_{t_0 + \Delta t}||_2$$

$$+ ||\boldsymbol{b}_{t_0 + \Delta t + 1} - \boldsymbol{b}'_{t_0 + \Delta t + 1}||_2 ||\boldsymbol{b}'^T_{t_0 + \Delta t}||_2 \bigg) \tag{66}$$

$$\leq ||\boldsymbol{A}_{t_0 + \Delta t} - \boldsymbol{A}'_{t_0 + \Delta t}||_2 + \epsilon_{\boldsymbol{b}, t_0 + \Delta t + 1} \alpha ||\boldsymbol{H}||^3 (2 b_{\max} + \epsilon_{\boldsymbol{b}, t_0 + \Delta t + 1}) \tag{67}$$

By induction, we find that the approximation error of $\boldsymbol{A}$ is quadratic in time for short times $0 \leq \Delta t \leq R/\alpha$,

$$||\boldsymbol{A}_{t_0+\Delta t} - \boldsymbol{A}'_{t_0+\Delta t}||_2 \leq \sum_{\widetilde{\Delta t}=0}^{\Delta t-1} \epsilon_{\boldsymbol{b},t_0+\widetilde{\Delta t}+1}\alpha||\boldsymbol{H}||^3(2b_{\max} + \epsilon_{\boldsymbol{b},t_0+\widetilde{\Delta t}+1}) \tag{68}$$

$$=||\boldsymbol{H}||^3 b_{\max}^2 \sum_{\widetilde{\Delta t}=0}^{\Delta t-1} \frac{\alpha^2}{2R}(\widetilde{\Delta t}+1)\left(2 + \frac{\alpha}{2R}(\widetilde{\Delta t}+1)\right) \tag{69}$$

$$\leq||\boldsymbol{H}||^3 b_{\max}^2 \sum_{\widetilde{\Delta t}=1}^{\Delta t} \frac{\alpha^2}{2R}\Delta t\left(2 + \frac{\alpha}{2R}\Delta t\right) \tag{70}$$

$$=||\boldsymbol{H}||^3 b_{\max}^2 \frac{\alpha^2 \Delta t^2}{2R}\left(2 + \frac{\alpha\Delta t}{2R}\right). \tag{71}$$

**Sixth part.** Now take $\Delta t = \lfloor r/\alpha \rfloor$, which for $r \to 0$ is eventually $\leq R/\alpha$ as required. As we sought to prove, the learning rate rescaled approximation error of the local drift direction and speed goes to zero:

$$\lim_{r\to 0}\lim_{\alpha\to 0} \frac{1}{r}||\boldsymbol{A}_{t_0+\lfloor r/\alpha\rfloor} - \boldsymbol{A}'_{t_0+\lfloor r/\alpha\rfloor}||_2 = \lim_{r\to 0}\lim_{\alpha\to 0} \frac{1}{r}||\boldsymbol{H}||^3 b_{\max}^2 \frac{\alpha^2\lfloor r/\alpha\rfloor^2}{2R}\left(2 + \frac{\alpha\lfloor r/\alpha\rfloor}{2R}\right) \tag{72}$$

$$= \lim_{r\to 0}||\boldsymbol{H}||^3 b_{\max}^2 \frac{r}{2R}\left(2 + \frac{r}{2R}\right) \tag{73}$$

$$= 0. \tag{32}$$

$\square$

## B  PROOF THAT $\mathbf{G}(\theta_t)\mathbf{H}$ HAS A NEGATIVE EIGENVALUE

This is true, because we can substitute $A = \mathbf{G}(\theta_t)$ and $B = \mathbf{H}$ in following lemma:

**Lemma 1.** *Let us $\boldsymbol{A}$ and $\boldsymbol{B}$ be symmetric, full-rank $n \times n$ matrices. Let $\boldsymbol{A}$ be positive-definite and $\boldsymbol{B}$ have at least one negative eigenvalue. Then, the product $\boldsymbol{AB}$ has at least one negative eigenvalue.*

*Proof.* Let $\boldsymbol{x}$ be an eigenvector of $\boldsymbol{B}$ with negative eigenvalue $\lambda$. Then, we have

$$\left(\boldsymbol{A}^{-1/2}\boldsymbol{x}\right)^T \boldsymbol{A}^{1/2}\boldsymbol{B}\boldsymbol{A}^{1/2}\left(\boldsymbol{A}^{-1/2}\boldsymbol{x}\right) \leq 0 \tag{74}$$

meaning that the positive-definite symmetric matrix $\boldsymbol{A}^{1/2}\boldsymbol{B}\boldsymbol{A}^{1/2}$ must have at least one negative eigenvalue $\lambda'$ with eigenvector $\boldsymbol{x}'$. Then, we have the eigenvalue equation

$$\boldsymbol{AB}\left(\boldsymbol{A}^{1/2}\boldsymbol{x}'\right) = \boldsymbol{A}^{1/2}\left(\boldsymbol{A}^{1/2}\boldsymbol{B}\boldsymbol{A}^{1/2}\right)\boldsymbol{x}' \tag{75}$$

$$= \lambda'\boldsymbol{A}^{1/2}\boldsymbol{x}' \tag{76}$$

which shows that $\boldsymbol{AB}$ has a negative eigenvalue $\lambda'$.  $\square$

## C  PROOF OF REPRESENTABILITY THEOREM

This section gives a proof of the representability theorem stated in Section 4.2:

**Theorem 1.** *Uniformly sample permutations $\boldsymbol{P}_i$ and create block-diagonal matrices $\boldsymbol{B}(\boldsymbol{\theta}^{(i)})$ where every block is $2 \times 2$, and whose block contents are listed by the parameters $\boldsymbol{\theta}^{(i)}$. Use these to construct the LODO subnetwork $\tilde{\boldsymbol{G}}(\boldsymbol{\theta})$ as in Equation 1 with some depth $N$ and hidden dimension $\tilde{n}$. Construct any linear neural network $\tilde{\boldsymbol{F}}$ with input dimension, output dimension, and number of*

*weights per layer at most $\tilde{n}$, at most $k$ incoming and at most $k$ outgoing weights for every neuron, depth $d$, and any arrangement of weights. Then, there is a probability of at least*

$$1 - \tilde{n}! N \sqrt{\frac{1}{2}\epsilon\left(\frac{\tilde{n}N}{4d(\lceil \log_2 k \rceil + 1)}, \tilde{n}\right)} \tag{11}$$

*that we can make $\tilde{G}(\boldsymbol{\theta}) = \tilde{F}$ for some $\boldsymbol{\theta}$.*

*Proof.* Our result comes in two parts: the first shows that $\tilde{G}(\boldsymbol{\theta})$ can represent arbitrary permutations, and the second shows that we can pick these $\tilde{G}(\boldsymbol{\theta})$ permutations and interleave them with block diagonal matrices to create a deeper $\tilde{G}(\boldsymbol{\theta})$ network which manifests any desired neural network. We use the terms fanin and fanout to mean the number of incoming and outgoing weights into and out of a neuron, respectively.

**First part.** Assume we would like to apply a given target permutation to $\tilde{n}$ elements using $\tilde{G}(\boldsymbol{\theta}) = \prod_{i=1}^{N} \boldsymbol{B}(\boldsymbol{\theta}^{(i)})\boldsymbol{P}_i$ consisting of $N$ layers with randomly chosen permutations $\boldsymbol{P}_i$. Our goal is to perform the target permutation given $\boldsymbol{P}_i$ by controlling the block diagonal matrices $\boldsymbol{B}(\boldsymbol{\theta}^{(i)})$. The form of $\tilde{G}(\boldsymbol{\theta})$ from Section 3 is

$$\tilde{G}(\boldsymbol{\theta}) = \prod_{i=1}^{N} \boldsymbol{B}(\boldsymbol{\theta}^{(i)})\boldsymbol{P}_i \tag{1}$$

which can be rewritten as

$$\tilde{G}(\boldsymbol{\theta}) = \boldsymbol{Q}_1 \prod_{i=1}^{N} \boldsymbol{Q}_i^T \boldsymbol{B}(\boldsymbol{\theta}^{(i)})\boldsymbol{Q}_i, \qquad \boldsymbol{Q}_N = \boldsymbol{P}_i, \quad \boldsymbol{Q}_{i-1} = \boldsymbol{P}_{i-1}\boldsymbol{Q}_i \tag{77}$$

with random independent uniform permutations $\boldsymbol{Q}_i$ instead. For each block in the matrix $\boldsymbol{B}(\boldsymbol{\theta}^{(i)})$, we may restrict ourselves to two options: to swap or to not swap the pair of indices. The conjugation by random permutation $\boldsymbol{Q}_i$ then shuffles these pairings, such that applying $\tilde{G}(\boldsymbol{\theta})$ is equivalent to repeatedly randomly pairing up indices for optional transposition instead, and then applying a final permutation $\boldsymbol{Q}_1$. We will work under this new equivalent formulation, since it is more amenable to analysis.

Let us choose to apply each transposition with probability $1/2$. Then, the expected entropy of $\tilde{G}(\boldsymbol{\theta})$ given the whole sequence of pairings is at least $\log \tilde{n}! - \epsilon(N\tilde{n}/2, \tilde{n})$ under Definition 1. In other words, the expected KL divergence of this distribution of $\tilde{G}(\boldsymbol{\theta})$ from the uniform is at most $\epsilon(N\tilde{n}/2, \tilde{n})$. Then by Pinsker's inequality, the expected total variation distance from the uniform is then at most

$$\sqrt{\frac{1}{2}\epsilon(N\tilde{n}/2, \tilde{n})}. \tag{78}$$

This guarantees that at most

$$\tilde{n}! \sqrt{\frac{1}{2}\epsilon(N\tilde{n}/2, \tilde{n})} \tag{79}$$

possible target permutations have a probability density of zero, in expectation, which is then an upper bound on the number of inaccessible target permutations. This means the probability that all target permutations are accessible is then at least

$$1 - \tilde{n}! \sqrt{\frac{1}{2}\epsilon(N\tilde{n}/2, \tilde{n})}. \tag{80}$$

Note that the leftmost $\boldsymbol{Q}_1$ in Equation 77 merely introduces a bijection between target permutations, and so does not change how many are accessible.

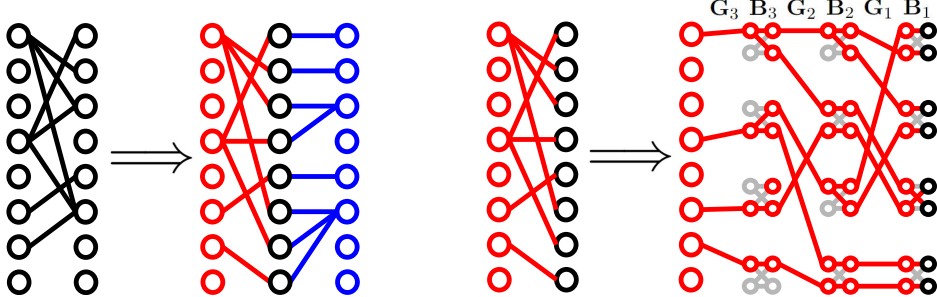

Figure 5: In this diagram, $\tilde{n} = 8$ and $p = 2$, for illustrative purposes. **Left:** Visual depiction of how a fanout forest followed by a fanin forest can represent arbitrary sparse connections. **Right:** Visual depiction of how $p + 1$ chained copies of $\tilde{G}(\boldsymbol{\theta})$ networks left-multiplied by block matrices diagonal can manifest a forest of fanout trees. $\tilde{G}_i$ are permutations implemented by $\tilde{G}(\boldsymbol{\theta})$ networks allowing for arbitrary connections, and $\boldsymbol{B}_i$ are block diagonal matrices which create the necessary fanouts. Data flows from left to right in the illustration, though successive operations are written out right to left when using mathematical notation. The largest fanout in this pattern of connections is 3, which is less than $2^p = 4$; all the fanins are 1.

**Second part.** Suppose we would like to represent $p$ target permutations using $p$ independently generated $\tilde{G}(\boldsymbol{\theta})$ networks each of depth $M$. Equation 80 lower bounds the probability that each network can represent its respective permutation. Then the probability that all of the $p$ target permutations are accessible by their respective copies of $\tilde{G}(\boldsymbol{\theta})$ is union bounded by

$$1 - p\tilde{n}!\sqrt{\frac{1}{2}\epsilon(M\tilde{n}/2, \tilde{n})} \tag{81}$$

where the union is over the probability that each copy fails to represent its target permutation. Given that this succeeds, we now need to chain these permutations together with block diagonal matrices to represent arbitrary neural networks $\tilde{\boldsymbol{F}}$, with Equation 81 lower bounding the probability of failure.

Suppose we are successful in representing any combination of $p$ target permutations using $p$ distinct independently generated $\tilde{G}(\boldsymbol{\theta})$ networks of depth $M$. Then, since each $\tilde{G}(\boldsymbol{\theta})$ network's final (left-most) operation is a block diagonal matrix, applying an additional block diagonal matrix afterward does not affect the operations that $\tilde{G}(\boldsymbol{\theta})$ can represent, since the set of block diagonal matrices we use is closed under matrix multiplication. Importantly, each block matrix can be used to copy a value from one index into two or to add two values together to leave one, creating fanin or fanout in the network. Then, interleaving $p + 1$ chained copies of $\tilde{G}(\boldsymbol{\theta})$ with block diagonal matrices left-multiplied for a total depth of $(p + 1)M$ therefore creates an aggregate operation which can still be represented by a single $\tilde{G}(\boldsymbol{\theta})$ network of depth $(p + 1)M$. This aggregate operation has up to $\tilde{n}$ arbitrary connections from input nodes to output nodes, with either all fanin at most 1 and all fanout at most $2^p$, or all fanout at most 1 and all fanin at most $2^p$. This is done by building a forest of fanin/fanout trees like illustrated on the right side of Figure 5.

If we compose such a $\tilde{G}(\boldsymbol{\theta})$ network of depth $(p + 1)M$ with fanout up to $2^p$ together with a $\tilde{G}(\boldsymbol{\theta})$ network of depth $(p + 1)M$ with fanin up to $2^p$, then we may represent any sparse matrix structure with at most $\tilde{n}$ nonzero weights and maximum fanin and fanout at most $2^p$, using a $\tilde{G}(\boldsymbol{\theta})$ network of depth $2(p + 1)M$. This construction is illustrated on the left side of Figure 5. We may adjust the final (leftmost) block diagonal matrix on the fanout side to change the values of the $\tilde{n}$ arbitrarily positioned weights in the desired sparse matrix. Therefore, any sparse matrix with at most $\tilde{n}$ weights and max node indegree and outdegree at most $k$ can be represented by a $\tilde{G}(\boldsymbol{\theta})$ of depth $2(\lceil \log_2 k \rceil + 1)M$.

Then, any linear neural network of depth at most $d$, at most $\tilde{n}$ weights per layer, and maximum fanin and fanout of at most $k$ can be represented by a $\tilde{G}(\boldsymbol{\theta})$ network of depth $2Md(\lceil \log_2 k \rceil + 1)$, by composition of sparse matrices. The probability that all of this is successful is merely the probability that all the $p = 2Md(\lceil \log_2 k \rceil + 1)$ permutations can be represented, which by Equation 81 is at

least

$$1 - 2\tilde{n}!Md(\lceil \log_2 k \rceil + 1)\sqrt{\frac{1}{2}\epsilon(M\tilde{n}/2,\tilde{n})}. \tag{82}$$

Thus in summary, if we randomly generate a $\tilde{G}(\boldsymbol{\theta})$ network of depth $N = 2Md(\lceil \log_2 k \rceil + 1)$ for fixed constants $k$, $d$, and $M$, $\tilde{n}$ neurons per layer, and block size $f = 2$, then there is a probability of at least

$$1 - \tilde{n}!N\sqrt{\frac{1}{2}\epsilon\left(M\tilde{n}/2,\tilde{n}\right)} = 1 - \tilde{n}!N\sqrt{\frac{1}{2}\epsilon\left(\frac{\tilde{n}N}{4d(\lceil \log_2 k \rceil + 1)},\tilde{n}\right)} \tag{83}$$

that $\tilde{G}(\boldsymbol{\theta})$ can represent every possible linear neural network of input and output dimension $\leq \tilde{n}$, depth $d$, at most $\tilde{n}$ nonzero weights per layer, and max fanin/fanout of at most $k$. $\quad\square$

## D  HESSIAN LEARNING LOCAL MINIMA (LODO CAN GENERALIZE)

Laurent & von Brecht (2018) gives a theorem showing that for dense linear neural networks on convex losses where each layer is at least as wide as the input or output layer, all local minima in the neural network weights are also global minima. If we simplify LODO by approximating the inverse Hessian with a full dense matrix $\boldsymbol{G}(\boldsymbol{\theta}_t)$, then this theorem applies to the Hessian approximation error $||\boldsymbol{I} - \boldsymbol{G}(\boldsymbol{\theta}_t)\boldsymbol{H}||_F^2$, which the rescaled error $||\boldsymbol{B}\boldsymbol{D}^{-1}||_F^2$ used in Section 4.1 is a proxy for. Thus we may expect that any inverse Hessian approximation which LODO could converge to is of similar high quality to the best possible inverse Hessian approximation.

## E  HYPERPARAMETERS

In every experiment, we tuned the hyperparameters of each optimizer using a genetic algorithm of 10 generations and 32 individuals per generation. Each hyperparamter was rescaled using $x \mapsto \ln x$ if the hyperparameter was a learning rate and $x \mapsto 1 - \ln(1 - x)$ if the hyperparameter was a decay parameter, so that the genetic algorithm would operate in a more well-behaved hyperparameter space. Starting from the default hyperparameters, each generation's mean hyperparameters were added to some Gaussian noise to create mutated hyperparameters for each individual, where the standard deviation of the noise was generation-dependent and followed a specific schedule. Each individual performed a generation-dependent number of steps of optimization, also according to a schedule. The next generation's mean hyperparameters were chosen to be the mean hyperparameters of the better performing half of the previous generation, as judged by average training loss during the last 10% of training. We also halved all the learning rates (equiv. initial learning rates for LODO versions) after tuning for the image generation task because tests showed the loss to diverge if training time horizons were longer than 8k steps. Since LODO is a random optimizer, we used a different randomization seed for every individual. Table 3 lists the parameters of the tuning schedule for every task. The tuned hyperparameters can be found in Table 4.

## F  TASK SETUP DETAILS

### F.1  NOISY QUADRATIC BOWL TASK DETAILS

This section fully explains the details of the setup of the noisy quadratic bowl task of Section 5.1. This 100 parameter task consists of a quadratic bowl for its loss landscape. Using a random uniform orthogonal matrix $\boldsymbol{U}$ and a diagonal matrix $\boldsymbol{D}$ consisting of a geometric sequence of 100 values starting at 0.001 and ending at 1, the Hessian of the quadratic bowl is set to $\boldsymbol{H} = \boldsymbol{U}\boldsymbol{D}\boldsymbol{U}^T$, and the center is set to the origin. However, whenever the loss and gradient are evaluated, the center of the quadratic bowl is perturbed by an i.i.d. random standard normal offset in each dimension. The initialization for this task is set to the origin. Due to the random wandering of the center, the expected loss rises linearly over time—unless the optimizer acts to prevent this, driving the error towards a steady state distribution. The expected loss after infinitely many steps can then be taken as a measure of the quality of an optimizer. The optimal solution for this task is to select the current

Table 3: Noise and step number schedules for tuning the optimizers' hyperparameters using the genetic algorithm presented in Appendix E

| Generation | Noisy Quadratic Bowl | | Rosenbrock Function | | Image Generation | |
|---|---|---|---|---|---|---|
| | Noise stddev | # steps | Noise stddev | # steps | Noise stddev | # steps |
| 0 | 3 | 1k | 3 | 200 | 3 | 1k |
| 1 | 3 | 1k | 3 | 200 | 3 | 1k |
| 2 | 3 | 1k | 3 | 200 | 3 | 1k |
| 3 | 3 | 1k | 2.5 | 200 | 3 | 1k |
| 4 | 2 | 1.5k | 2 | 200 | 2 | 1.5k |
| 5 | 1.7 | 1.5k | 1.5 | 200 | 1.7 | 1.5k |
| 6 | 1.4 | 2k | 1 | 200 | 1.4 | 2k |
| 7 | 1.2 | 3k | 0.75 | 200 | 1.2 | 3k |
| 8 | 0.9 | 5k | 0.5 | 200 | 0.9 | 5k |
| 9 | 0.6 | 8k | 0.3 | 200 | 0.6 | 8k |

minimum of the quadratic bowl at every timestep (which is what the Newton method would do). Due to the movement of the minimum between steps and loss evaluations, we should still expect this strategy to a achieve nonzero loss which can be analytically calculated to be 7.412. Table 5 shows various optimizers' performances after having taken many steps.

### F.2 CNN Image Generation Task Details

This section explains the CNN image generation task of Section 5.2, which is similar to the task of training a PixelCNN (Oord et al., 2016). Like PixelCNN, our CNN generates pixels row by row and column by column, and classifies the brightness of each pixel into 256 classes with crossentropy loss. Our data preprocessing is as follows. An MNIST image randomly selected, and one pixel location is chosen uniformly at random and blackened. All pixels below, or to the right of and in the same row as the selected pixel are blackened. The input into the CNN consists of this partially masked/blackened image (divided by 256 for normalization), an image indicating which pixels are specifically masked/blackened (indicated by $-1$ and 1), another image indicating which pixels are left of the selected pixel (indicated by $-1$ and 1), an image of a linear gradient from $-1$ to 1 in the x direction, and the same for the y direction. The last three images are present purely to break horizontal and vertical translation symmetries in the data, which has been shown to be helpful in vision tasks involving collection of information from specific locations of an image specified by the data (Liu et al., 2018). The preprocessed data is visualized in Figure 6.

The CNN architecture we use for autoregression consists of:

- 5 input channels as previously described.
- Residual connection to Point A.
  - One pixel of zero padding, and convolution with 3 by 3 filters to 20 channels, with no activation.
- Point A.
- The following indented items are repeated 5 times.
  - The following indented items are repeated 4 times.
    * Residual connection to Point B.
      · Convolution with 1 by 1 filters to 40 channels, with arctangent activation. We use the arctangent to mitigate gradient norm issues.
      · One pixel of zero padding, and three different depthwise convolutions concatenated together, with 3 by 3 filters to 120 channels, with arctangent activation.
      · Convolution with 1 by 1 filters to 20 channels, with no activation.
    * Point B.

Table 4: Hyperparameters used for the experiments in Section 5, after tuning hyperparameters with a genetic algorithm as in Appendix E and halving the learning rates (equiv. initial learning rates for LODO variants) for the image generation task. $\beta$ and $\beta_1$ generally represent momentum decay rates while $\beta_2$ represent variance EMA decay rates. Dashes indicate that the optimizer was not used for that experiment.

| Optimizer | Hyperparameter | Value | | |
| --- | --- | --- | --- | --- |
| | | Noisy quadratic bowl | Rosenbrock function | Image generation |
| Adam | Learning Rate | 1.164 | 0.9704 | 0.0009554 |
| | $\beta_1$ | 0.465 | 0.864 | 0.9323 |
| | $\beta_2$ | 0.9884 | 0.99804 | 0.99505 |
| Momentum | Learning Rate | 1.394 | 0.09870 | 0.003411 |
| | $\beta$ | 0.529 | 0.9359 | 0.9640 |
| RMSprop | Learning Rate | 0.449 | 0.004318 | $4.167 \times 10^{-5}$ |
| | $\rho$ | 0.04943 | 0.02836 | 0.002258 |
| | $\beta$ | 0.595 | 0.880 | 0.936 |
| Yogi | Learning Rate | 2.169 | 0.2991 | 0.0009340 |
| | $\beta_1$ | 0.4362 | 0.9273 | 0.9319 |
| | $\beta_2$ | 0.9999723 | 0.998787 | 0.999708 |
| LARS | Learning Rate | – | – | 0.0008552 |
| | $\beta$ | – | – | 0.9343 |
| | Weight Decay | – | – | $4.839 \times 10^{-5}$ |
| L-BFGS | Learning Rate | 1.204 | – | – |
| | $\tau$ | 23002 | – | – |
| O-LBFGS | Learning Rate | 1.050 | – | – |
| | $\tau$ | 36721 | – | – |
| LODO | Meta-Learning Rate | 0.009600 | 0.0001394 | $7.946 \times 10^{-6}$ |
| | $\beta$ | 0.195 | 0.897 | 0.9343 |
| | Initial Learning Rate | 0.270 | 0.2946 | 0.08459 |
| LODO-Diagonal | Meta-Learning Rate | – | – | 0.0005196 |
| | $\beta$ | – | – | 0.9860 |
| | Initial Learning Rate | – | – | 0.1325 |
| LODO-Global | Meta-Learning Rate | – | – | $7.951 \times 10^{-5}$ |
| | $\beta$ | – | – | 0.9525 |
| | Initial Learning Rate | – | – | 0.05583 |
| LODO-Residuals | Meta-Learning Rate | – | – | $3.829 \times 10^{-5}$ |
| | $\beta$ | – | – | 0.9301 |
| | Initial Learning Rate | – | – | 0.02134 |
| LODO-SGD | Meta-Learning Rate | – | – | 0.0007196 |
| | $\beta$ | – | – | 0.9499 |
| | Initial Learning Rate | – | – | 0.1035 |

> – Average pooling of 2 by 2 blocks to halve the image size, with a pixel of zero padding beforehand if the image size is odd.

- Convolution with 1 by 1 filters to 256 channels, with softmax activation.

The loss is then the average crossentropy between true brightness of the query pixel and the distribution given by the softmax output of this CNN, over a batch size of 256 images. The whole CNN has about 94696 parameters, which are initialized with LeCun normal initialization (Klambauer et al., 2017). The task for the optimizer is to train these parameters to minimize this loss.

Figure 7 shows some imitation MNIST images generated using this CNN by sampling pixels one by one.

Table 5: Average tracking error of the quadratic bowl minimum on the noisy quadratic bowl task of Section 5.1. Values are are averaged over the last 10% of training before the stated training milestone, with means and standard deviations produced across 8 optimization runs with different randomization seeds. The theoretical best possible loss using Newton's method is also listed. The version of L-BFGS is one with stochastic modifications from (Schraudolph et al., 2007), instead of the original from (Nocedal & Wright, 1999). Our timing setup is described in Appendix H.

| | Training loss | |
|---|---|---|
| Optimizer | 100k steps | 300 sec. |
| Adam (Kingma & Ba, 2014) | $15.28 \pm 0.07$ | $15.26 \pm 0.08$ |
| Momentum | $16.59 \pm 0.08$ | $16.56 \pm 0.05$ |
| RMSprop (Hinton et al., 2012) | $22.37 \pm 0.06$ | $22.47 \pm 0.13$ |
| Yogi (Zaheer et al., 2018) | $15.35 \pm 0.10$ | $15.16 \pm 0.05$ |
| L-BFGS (Schraudolph et al., 2007) | $49.30 \pm 1.04$ | $40.60 \pm 1.40$ |
| O-LBFGS (Schraudolph et al., 2007) | $13.96 \pm 0.08$ | $10.80 \pm 0.17$ |
| **LODO (ours)** | $\mathbf{8.99 \pm 0.05}$ | $\mathbf{10.05 \pm 0.22}$ |
| Newton Method (Best Possible) | 7.41 | |

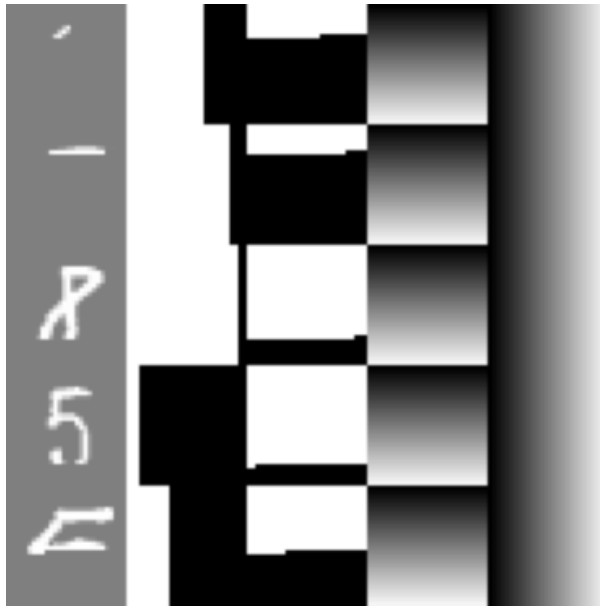

Figure 6: Batch of 5 tensors of preprocessed MNIST images from the image generation task of Section 5.2, each as one of the 5 rows of the image. Shown in each of 5 columns are the masked input, the left-of-selected-pixel indicator, visible mask, and two gradient images. The images in each row are concatenated together into a $28 \times 28 \times 5$ data tensor and then used as input into the CNN.

## G ROSENBROCK FUNCTION MINIMIZATION EXPERIMENT

We probe the behavior of LODO with a small test task of finding the minimum of a rescaled Rosenbrock function $f(x, y) = 0.01(x - 1)^2 + (x^2 - y)^2$, which has no local minima and one global minimum at $(x, y) = (1, 1)$. We initialized the optimizers at $(x, y) = (-0.5, 2)$ and gave them 200 steps to run. The trajectory taken by LODO, shown in Figure 9, is similar to the short timescale dynamics of other optimizers using momentum, in that it tends to overshoot and then correct itself in an oscillatory manner. Learning curves in Figure 8 and losses in Table 6 show the performance of all the optimizers on this task.

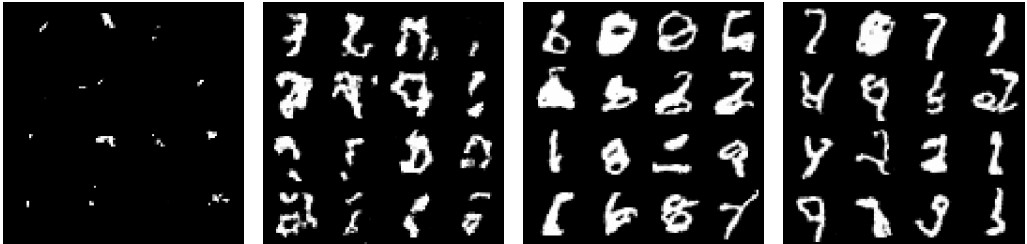

Figure 7: Grids of 16 imitated MNIST images generated the image generation CNN of Section 5.2, trained for 300k steps using RMSprop, Adam, Yogi, and LODO respectively.

Table 6: Mean losses between steps 180-200 while training on the Rosenbrock function minimization task, with various optimizers. The standard deviation is calculated over 8 test runs of LODO because its $G$ neural network architecture is randomly generated.

| Optimizer | Mean loss between steps 180-200 |
|---|---|
| Adam | 0.00005342 |
| RMSprop | 0.0008967 |
| Momentum | 0.01397 |
| Yogi | 0.0007916 |
| LODO (ours) | $0.001040 \pm 0.00002140$ |

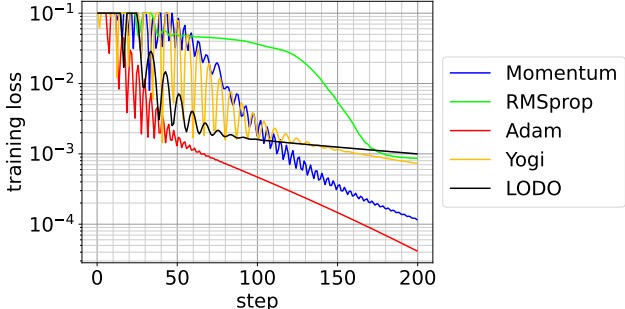

Figure 8: Log loss as a function of step, when using various optimizers on the Rosenbrock function minimization task.

## H  TRAINING LOOP TIMING DETAILS

This section describes how we timed each optimizer to report performance at specified times and step per second training speeds. We performed all optimization runs in TensorFlow 2, each with 40 Intel Xeon Gold 6248 CPUs and 2 Nvidia Volta V10 GPUs. Time reported includes all training time (forward and backward propagation, optimizer computation, data loading, preprocessing, etc.) except it does not include time taken to evaluate metrics such as the Hessian approximation error and the validation and test losses and accuracies.

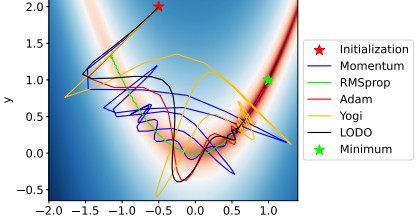

Figure 9: 100 step trajectories of various optimizers on the Rosenbrock function minimization task. The red star marks the initialization and the green star marks the location of the global minimum.

