# OpenReview forum: "Learning to Optimize Quasi-Newton Methods"
_ICLR.cc/2023/Conference — Submitted to ICLR 2023_

### Official Review · Reviewer_oyKA · 2022-10-24

**Confidence:** 4
**Correctness:** 2
**Technical Novelty And Significance:** 2
**Empirical Novelty And Significance:** 2
**Recommendation:** 3

**Clarity, Quality, Novelty And Reproducibility:**

The paper is clear.
Novelty is questionable given that many other methods meta-learn a pre-conditioner for gradient descent, some of which are not cited / compared with.


**Strength And Weaknesses:**

Strengths

- Meta-learning optimizers is a promising area of research.


Weaknesses

Some of the claims in this paper are wrong or obvious, and comparison with previous work is not appropriate.

- Section 4.1 is trivial. It is obvious that the method learns the inverse Hessian in the case of a quadratic loss, because it is known to be the best pre-conditioner in that case (it converges in one step from any initial condition).
However, for non-quadratic loss functions the best pre-conditioner is not the inverse Hessian, and the proposed method will not learn the inverse Hessian.
This is particularly obvious for non-convex loss functions, where the inverse Hessian is not even positive definite and Newton's method would not even converge to a minimum of the loss (see e.g. https://arxiv.org/abs/1406.2572)

- The statement "the true inverse Hessian is guaranteed to be symmetric positive-definite" is wrong, that is not true for non-convex losses.
I assume that is the field of application of this method since the authors apply their method to non-convex losses in sections 5.2, 5.3.
If the authors refer to convex losses, then they should compare their method with other methods for convex optimization.

- Related to the last point, for autoregressive image generation the authors report that "we could not get any quasi-Newton methods to converge on this task."
It seems that authors are not aware that quasi-newton methods should not be expected to converge on non-convex problems, again because the Hessian is not positive definite.

- Comparison with other methods is underwhelming and other, more powerful methods are not even considered.
I do not even consider the quadratic problem because results in that case are trivial.
For the autoregressive image generation task, the proposed method does not perform significantly better than momentum.
Other, more powerful methods are not even compared with (see e.g. https://arxiv.org/abs/2006.08877).

- Even without all problems above, the method is not so novel.
Many other studies have tried to learn pre-conditioner for neural networks, inclusing some that the authors do not cite (see e.g. https://arxiv.org/abs/1902.03356)


Minor:
- Even in the quadratic case, in Fig.3 (Left) it seems that the error does not converge to zero, meaning that the method fails to learn the inverse Hessian.


**Summary Of The Paper:**

This paper proposes to meta-learn an optimizer to approximate Newton's method, namely to learn a pre-conditioner for gradient descent to approximate the inverse Hessian.
It proves that the method learns the inverse Hessian for quadratic loss functions, and that the proposed parameterization of the pre-conditioner is expressive under some circumstances.
The method is tested on autoregressive image generation for MNIST.



**Summary Of The Review:**

The problem is interesting but some of the claims in this paper are wrong or obvious, and comparison with previous work is not appropriate.

---

> ### Author Response · Authors · 2022-11-14
> **Specific Responses to Reviewer oyKA Continued**
>
> ### Novelty
>
> > Even without all the problems above, the method is not so novel. Many other studies have tried to learn pre-conditioner for neural networks, including some that the authors do not cite (see e.g. https://arxiv.org/abs/1902.03356)
>
> Thank you for providing this reference which we missed; we have added it into our Related Work. We agree that the idea of learning/estimating pre-conditioners is already well-studied—after all this is a fundamental idea which drives general approaches to optimization including few step fine-tuning, L2O, quasi-Newton methods, and natural gradients. This was not an idea we sought to introduce though; fundamentally our contributions build on this existing idea by exploring two more novel ideas, namely the following (boiled down):
>
> 1. We can learn this preconditioner online without meta-training time nor a task distribution, an idea which is rarely found if at all in L2O literature; Reviewer **XFBK** states that
>
>     > The construction of LODO seems interesting, which could inspire future development on online L2O methods without any meta-training.
>
>     and Reviewer **w1wa** states that
>
>     > The whole idea is quite attractive since it can lead to algorithms that will take advantage of Newton-type methods using the strengths of the L2O framework.
>
>     The method in the paper you linked (Park and Oliva, 2019, https://arxiv.org/abs/1902.03356) "focus[es] on a fast adaptation scenario requiring a small number of gradient steps" rather than the construction of "a generic optimizer that is broadly applicable for different datasets and architectures"—what LODO and L2O in general does.
>
> 2. Our parameterization of the preconditioner is almost always able to wholly represent a huge class of other parameterizations smaller by a logarithmic factor in parameter count (with some restrictions). Furthermore we can adjust this parameter count over a wide range by modifying the depth $N$ of $\tilde{G}(θ)$, without breaking the aforementioned property. In this sense, LODO has an optimally parameter-efficient parameterization of the preconditioner up to a logarithmic factor, while other methods such as the paper you mentioned (Park and Oliva, 2019, https://arxiv.org/abs/1902.03356) —while powerful—have no such properties.

---

> ### Author Response · Authors · 2022-11-14
> **Specific Responses to Reviewer oyKA**
>
> ### Triviality of the Noisy Quadratic Bowl Setup
>
> > Section 4.1 is trivial. It is obvious that the method learns the inverse Hessian in the case of a quadratic loss, because it is known to be the best pre-conditioner in that case (it converges in one step from any initial condition).
>
> We respectfully disagree that Section 4.1 presents trivial or obvious results. It is not clear from the statement of Algorithm 1 (which contains no explicit Hessian anywhere) that LODO would correctly learn the inverse Hessian as opposed to diverging or learning a different preconditioner, especially since the task is stochastic enough to cause L-BFGS and O-LBFGS to diverge in practice in Section 5.1. It would be ill-informed of us to assume that our algorithm performed as we designed it to without any evidence, and it is more principled for us to start with theory which is as close to a proof of correctness as possible. Furthermore, we have uncovered some nontrivial results, such as that LODO learns the Hessian *through a combination of noise and curvature*, unlike many other quasi-Newton methods.
>
> > I do not even consider the quadratic problem because results in that case are trivial.
>
> We also respectfully disagree that the results of the quadratic problem are trivial, even given that it is a test of the theory of Section 4.1. For one, we have used many approximations and limits in Section 4.1, and though we provide justification for these, we still require the experiment of Section 5.1 to verify that they suffice in practice. The two most notable approximations used are that $G(θ)$ is parameterized by the individual entries in a dense matrix while in the experiment $G(θ)$ is parameterized as in Equation 1; and that the momentum is zero, while in the experiment the momentum is not. It is not obvious that either of these approximations does not nullify the efficacy of LODO in practice, and the experiment in Section 5.1 shows these approximations to be of value.
>
> ### Applicability to Non-convex Loss Landscapes
>
> > The statement "the true inverse Hessian is guaranteed to be symmetric positive-definite" is wrong, that is not true for non-convex losses. I assume that is the field of application of this method since the authors apply their method to non-convex losses in sections 5.2, 5.3. If the authors refer to convex losses, then they should compare their method with other methods for convex optimization.
>
> Good catch! What we really meant to say is that we design LODO such that the inverse Hessian estimate $G(θ)$ is positive-definite, to guarantee that the step has a positive dot product with the gradient. This helps LODO navigate saddle points. We have updated the statement accordingly, and have also included an analysis of the non-convex case (see our general response).
>
> > Related to the last point, for autoregressive image generation the authors report that "we could not get any quasi-Newton methods to converge on this task." It seems that authors are not aware that quasi-newton methods should not be expected to converge on non-convex problems, again because the Hessian is not positive definite.
>
> In fact, we exactly agree that most quasi-Newton methods should not be expected to converge on non-convex problems. This is precisely why we tested other quasi-Newton methods, to motivate that LODO's capabilities extend into the non-convex realm beyond where many other quasi-Newton methods become ineffective, despite LODO being a quasi-Newton method itself.

---

> ### Author Response · Authors · 2022-11-17
> **We Look Forward to Your Feedback**
>
> Dear Reviewer **oyKA**,
>
> We would greatly appreciate it if you would let us know whether we have adequately addressed your concerns, as well as if you have any further feedback. In our general response and specific response to you, we:
>
>  - highlight our contributions and nontrivial results found in the noisy quadratic bowl setup;
>  - analyze the non-convex/non-PSD Hessian case and discuss motivations for the experiments we perform in this setting;
>  - explain that LODO's performance indicates its ability to bring L2O and quasi-Newton methodology to the meta-training-free nonconvex setting, where other L2O and quasi-Newton methods are inapplicable; and
>  - highlight the novel components of our paper.
>
> We look forward to hearing your feedback!

---

> ### Author Response · Authors · 2022-12-09
> **Has our response addressed your concerns?**
>
> Thank you for your constructive feedback on our paper, it has allowed us to improve our paper. More specifically, we have added to our theory and ablations, alleviated potential misunderstandings of the novelty of our work to the background literature, clarified our writing, and restructured our paper's sections.
>
> Best,
>
> The Authors

---

### Official Review · Reviewer_XFBK · 2022-10-24

**Confidence:** 3
**Correctness:** 3
**Technical Novelty And Significance:** 3
**Empirical Novelty And Significance:** 2
**Recommendation:** 5

**Clarity, Quality, Novelty And Reproducibility:**

The writing is generally good. Please specify the requirement for the Hessian matrix $H$ in section 4.1, and reflect it in the proof.

**Strength And Weaknesses:**

Strength:
- The construction of LODO seems interesting, which could inspire future development on online L2O methods without any meta-training.
- The theoretical discussion on the inverse Hessian approximation and the expressiveness seems solid.

Weaknesses:
- The empirical performance of LODO is not promising, and only limited experimental results are provided.
- I believe that the theoretic discussion in section 4.1 requires the Hessian $H$ to be at least PSD, since $G(\theta_t)$ is constructed to be PSD. But it is not clear to me which step of the proof relies on the PSDness. This needs to be clarified.
- When the objective function is non-convex, such that $H$ is potentially not PSD, what will be learned by $G(\theta_t)$? Will it capture all the positive curvature of the loss landscape and neglect the negative curvature?
- I think more ablation study for the effect of each algorithmic component in LODO is necessary, such as comparing LODO without momentum, LODO, plain momentum, LODO using SGD to tune the model, etc.

-----------------------------------
Post rebuttal: Thank the authors for the response and the additional experiments, which resolve some of my concerns. However, I still think the empirical performance of LODO is not strong enough, and I agree with reviewer oyKA that the comparison with other methods is underwhelming. Thus, I keep my assessment.

**Summary Of The Paper:**

This work proposes LODO, an L2O that performs quasi-Newton optimization without any meta-training. Specifically, LODO has a hypergradient optimization structure while the parameterization of the approximated inverse Hessian is chosen to be a linear neural network. The authors prove that under certain simplified situation, LODO learns the true inverse Hessian, and that the linear neural network construction is expressive in that it can represent all linear neural networks smaller by a logarithmic factor. Experiments are provided to evaluate the effectiveness of LODO.

**Summary Of The Review:**

This work can be seen as an initial attempt for developing online L2O methods without any meta-training. However, the discussion of LODO in the current version is insufficient, see the main review.

---

> ### Author Response · Authors · 2022-11-14
> **Specific Responses to Reviewer XFBK**
>
> ### Diversity of Ablations
>
> > I think more ablation study for the effect of each algorithmic component in LODO is necessary, such as comparing LODO without momentum, LODO, plain momentum, LODO using SGD to tune the model, etc.
>
> Thank you for the helpful suggestion. We have added two ablations to Table 3: LODO without momentum and LODO using SGD to tune the model, as well as explanations of these ablations at the beginning of the list of ablations as well as within the Effects of EMAs subsubsection. All of these perform worse than LODO. We have also updated Table 5 in the Appendix to include hyperparameter details for these ablations as well.

---

> ### Author Response · Authors · 2022-11-17
> **We Look Forward to Your Feedback**
>
> Dear Reviewer **XFBK**,
>
> We would greatly appreciate it if you would let us know whether we have adequately addressed your concerns, as well as if you have any further feedback. In our general response and specific response to you, we:
>
>  - explain that LODO's performance indicates its ability to bring L2O and quasi-Newton methodology to the meta-training-free nonconvex setting, where other L2O and quasi-Newton methods are inapplicable;
>  - analyze the non-convex/non-PSD Hessian case; and
>  - add more ablation experiments.
>
> We look forward to hearing your feedback!

---

> ### Author Response · Authors · 2022-12-09
> **Has our response addressed your concerns?**
>
> Thank you for your constructive feedback on our paper, it has allowed us to improve our paper. More specifically, we have added to our theory and ablations, alleviated potential misunderstandings of the novelty of our work to the background literature, clarified our writing, and restructured our paper's sections.
>
> Best,
>
> The Authors

---

### Official Review · Reviewer_w1wa · 2022-10-27

**Confidence:** 4
**Correctness:** 4
**Technical Novelty And Significance:** 3
**Empirical Novelty And Significance:** 2
**Recommendation:** 6

**Clarity, Quality, Novelty And Reproducibility:**

The paper presents novel and interesting ideas. However, Section 4, which presents theoretical results, is not clearly written making it hard to follow the importance of the results. Also, the experimental section could be improved. It would be better to move some experiments to the appendix rather than keeping all of them in the main paper but leaving important details of them in the appendix due to space limitations.

**Strength And Weaknesses:**

Strengths:

1) Learn-to-optimize (L2O) algorithms leverage the power of deep neural networks to achieve better convergence rates for several problems over classical optimization algorithms.
2) LODO builds on these ideas to come up with a novel L2O approach for quasi-Newton algorithms.
3) The whole idea is quite attractive since it can lead to algorithms that will take advantage of Newton-type methods using the strengths of the L2O framework.

Main comments/ Weaknesses:
1) It is hard to follow several parts of the paper. Specifically, Section 4 is poorly written. In Sect. 4.1 the authors describe the Hessian learning dynamics without giving the full details of the setup they consider. Hence it is hard to follow the ideas given later.
2) In Section 4.2, Definition 1, the definition of function $\epsilon$ is missing. Besides, it is hard to follow the text, which is densely written and not clear at all.
3) The statement of Theorem 1 lacks clarity and important details are missing. How $G(\theta)$ is sampled ? What is M? Also $\epsilon()$ is not defined (see also above).
4) In Section 5, Section 5.1., it is not clear why the training loss of other methods .e.g. RMSprop, Momentum etc. remains so much higher than LODO.
5) Section 6 could merge with Section 5.




**Summary Of The Paper:**

In this paper, the authors introduce a learn-to-optimize (L20)-based approach for quasi-Newton algorithms. The main idea is to learn on-the-fly an approximation of the inverse Hessian matrix modeled as the product of random permutation and block diagonal matrices. The proposed architecture called LODO consists of a linear neural network whose depth affects the representation power of the Hessian approximate. The authors provide a theoretical representability theorem showing that LODO can exactly represent the Hessian matix with high probability that increases with the depth of the linear neural network. They also provide simulated and an MNIST image generation experiment showing the promising performance of the proposed approach.

**Summary Of The Review:**

The  paper introduces a L2O-based algorithm for quasi-Newton methods. The idea of approximating the inverse Hessian with linear neural networks whose parameters are learned on-the-fly is interesting. However, theoretical results are not presented clearly. Hence, it is hard to assess their significance and value the importance of the specific parametrization of the inverse Hessian introduced in the paper.
Moreover, the organization of the experimental part could be improved.

-----------------------------------------------------
Post-Rebuttal Comment:
I would like to thank the authors for their time and effort to address my comments/concerns.
At this time, I will my score unchanged.

---

> ### Author Response · Authors · 2022-11-14
> **Specific Responses to Reviewer w1wa**
>
> ### Readability of Theoretical Sections
>
> > The paper presents novel and interesting ideas. However, Section 4, which presents theoretical results, is not clearly written making it hard to follow the importance of the results.
>
> > It is hard to follow several parts of the paper. Specifically, Section 4 is poorly written. In Sect. 4.1 the authors describe the Hessian learning dynamics without giving the full details of the setup they consider. Hence it is hard to follow the ideas given later.
>
> > In Section 4.2, Definition 1, the definition of function $\epsilon$ is missing. Besides, it is hard to follow the text, which is densely written and not clear at all.
>
> > The statement of Theorem 1 lacks clarity and important details are missing. How $G(\theta)$ is sampled? What is $M$? Also $\epsilon()$ is not defined (see also above).
>
> Thank you for pointing this out, we have clarified Section 4.1 by including a summary of the full problem setup, algorithm, and conclusion at the beginning of Section 4.1; and we have rewritten Section 4.2 in a more conceptually clear fashion while trying to maintain technical precision.
>
> Definition 1 is intended to define $\epsilon(N\tilde{n}/2,\tilde{n})$ as the mean information which remains in a composite permutation after an identity permutation of $\tilde{n}$ elements undergoes $N$ iterations of randomly pairing the elements and swapping each with probability half. We reworded Definition 1 to make this more clear.
>
> We intended to refer readers to Equation 1 of Section 3 for the method of sampling $G(\theta)$. To make the sampling method more clear, we have now given explicit instructions for this in Theorem 1. $M$ is a quantity related to neural network depth, utilized in the second part of the proof of Theorem 1 in the Appendix; thank you for helping us realize that Theorem 1 can be simplified by rephrasing it in a way which does not need to mention $M$. We have rewritten the theorem and results which follow accordingly. To be clear on $M$, it represents the number of layers of $\tilde{G}(\theta)$ we compose to allow this block of layers to express arbitrary permutations—the expression of many such arbitrary permutations plays a key role in proving that $\tilde{G}(\theta)$ as a whole is highly expressive.
>
> We hope that our clarification as listed above and in our general response helps to better convey our theoretical results: Section 4.1 on the training dynamics and the role of noise when learning the inverse Hessian, and Section 4.2 on the optimality of our choice of inverse Hessian representation.
>
> ### Structure and Formatting
>
> > Section 6 could merge with Section 5.
>
> Thank you for pointing this out. We agree, and have moved Section 6.x to Section 5.3.x for all x, since all the ablations of Section 6 are modifications of the image generation experiment in Section 5.3.
>
> > Also, the experimental section could be improved. It would be better to move some experiments to the appendix rather than keeping all of them in the main paper but leaving important details of them in the appendix due to space limitations.
>
> This is a good suggestion. We have moved the Rosenbrock function experiment to the Appendix. We have also moved the table of noisy quadratic bowl experiment results to the Appendix to accomodate for added text.

---

> ### Author Response · Authors · 2022-11-17
> **We Look Forward to Your Feedback**
>
> Dear Reviewer **w1wa**,
>
> We would greatly appreciate it if you would let us know whether we have adequately addressed your concerns, as well as any further feedback you might have. In our general response and specific response to you, we:
>
>  - clarify the setup of the noisy quadratic bowl;
>  - clarify our theoretical section regarding inverse Hessian representability, especially the definition of $\epsilon$ and Theorem 1;
>  - explain the worse performance of other optimizers in the noisy quadratic bowl experiment; and
>  - merge the experiment and ablation sections together and move some experiments to the appendix.
>
> We look forward to hearing your feedback!

---

> ### Author Response · Authors · 2022-12-09
> **Has our response addressed your concerns?**
>
> Thank you for your constructive feedback on our paper, it has allowed us to improve our paper. More specifically, we have added to our theory and ablations, alleviated potential misunderstandings of the novelty of our work to the background literature, clarified our writing, and restructured our paper's sections.
>
> Best,
>
> The Authors

---

### Author Response · Authors · 2022-11-14
**General Responses to the Reviewers Continued**

### Empirical Performance

Reviewer **XFBK**:
> The empirical performance of LODO is not promising, and only limited experimental results are provided.

Reviewer **oyKA**:
> Comparison with other methods is underwhelming and other, more powerful methods are not even considered.

> For the autoregressive image generation task, the proposed method does not perform significantly better than momentum. Other, more powerful methods are not even compared with (see e.g. https://arxiv.org/abs/2006.08877).

We would like to point out that L2O methods are not usually even applicable without meta training, while Reviewer **oyKA** notes that

> quasi-newton methods should not be expected to converge on non-convex problems,

and yet LODO—a quasi-Newton L2O method—performs competitively in the non-convex image generation task without any meta training. Thus, the competitiveness of LODO gestures towards the extension of quasi-Newton and L2O methodology towards more modern machine learning optimization settings, where most of the related methods tend to be futile. LODO's performance has convinced Reviewer **w1wa** to summarize experiments in the paper by writing that

> They also provide simulated and an MNIST image generation experiment showing the promising performance of the proposed approach.

As stated in the second paragraph of the introduction, we specifically target the Hessian; we do not learn other preconditioners such as the inverse Fisher information matrix for example, making our method quasi-Newton and not natural gradient instead. We make no claim that the Hessian nor any modification of it (eg. constrained to be PSD such as in LODO) is the best possible gradient preconditioner leading to faster convergence in a more realistic setting such as the image generation task in Section 5.3; we leave this question for future work.

To the best of our knowledge, all other known powerful gradient descent preconditioner parameterizations such as the paper mentioned by Reviewer **oyKA** (Goldfarb et al., 2020, https://arxiv.org/abs/2006.08877) rely on known structure of the parameter space in relation to a specific application-dependent neural network architecture to be trained. We designed LODO to be architecture agnostic, so we feel it is unfair to compare these other methods to LODO. Nonetheless, we included a reference to this paper in our Related Work.

### Preconditioner Quality in the Noisy Quadratic Bowl Experiment

Reviewer **w1wa**:
> In Section 5, Section 5.1., it is not clear why the training loss of other methods. e.g. RMSprop, Momentum etc. remains so much higher than LODO.

Reviewer **oyKA**:
> Even in the quadratic case, in Fig.3 (Left) it seems that the error does not converge to zero, meaning that the method fails to learn the inverse Hessian.

The training loss and Hessian approximation errors are both measures of the quality of the preconditioner used. LODO is better at approximating the true Hessian and so performs better in this task than the other methods.

Our theory in Section 4.1 that the method learns the inverse Hessian to perfection is contingent on an assumption that $G(θ)$ is parameterized elementwise and densely, whereas the experiment is run with $G(θ)$ parameterized as in Equation 1 instead. Ordinarily, our theory in Section 4.2 would motivate that our parameterization is expressive enough to account for this, but it requires that there is a logarithmic penalty in free parameter count, which outstrips the 1.27x gap in free parameter count in our experiment setup. This is responsible for the failure of LODO to exactly replicate the inverse Hessian.

Despite this, LODO's preconditioner gets much closer to the true inverse Hessian than other methods' preconditioners. Adam, Momentum, RMSprop, and Yogi implicitly precondition their gradients with diagonal matrices where most of the true Hessian's Frobenius norm resides off the diagonal, and thus they achieve expected approximation errors of $\sigma$ very close to 1. Similarly, L-BFGS and O-LBFGS precondition their gradients with low rank matrices and thus cannot suppress most of the true Hessian which is high rank with many large singular values, therefore their expected approximation errors are also very close to 1. In this context, LODO achieves a much better approximation of the inverse Hessian (with error $\sigma\approx0.75$) than all the other methods we test. We included an elaboration on this point at the end of Section 5.1, that all these optimizers create and use a gradient preconditioner, with LODO's being the closest to the true Hessian and thus leading to the lowest empirical loss.

#### Correction to Hyperparameter Table in Appendix
In addition to changes in response to comments by the Reviewers, we have also corrected some clerical errors in Table 5 of the Appendix, which lists optimizer hyperparameters.

---

### Author Response · Authors · 2022-11-14
**General Responses to the Reviewers**

### We Thank the Reviewers

We thank the Reviewers for providing constructive criticism for our paper, as this has helped us to improve it. We have carefully considered each suggestion and criticism the Reviewers made, and we have given a detailed response for each, with any changes highlighted in blue in the revised version. We welcome the Reviewers to provide more feedback to help us further improve our paper. We also kindly ask the Reviewers to consider increasing their scores in light of our responses to their concerns.

### Behavior Near Saddle Points

Reviewer **XFBK**:
> I believe that the theoretic discussion in section 4.1 requires the Hessian $H$ to be at least PSD, since $G(θ_t)$ is constructed to be PSD. But it is not clear to me which step of the proof relies on the PSDness. This needs to be clarified.

> When the objective function is non-convex, such that $H$ is potentially not PSD, what will be learned by $G(θ_t)$? Will it capture all the positive curvature of the loss landscape and neglect the negative curvature?

> Please specify the requirement for the Hessian matrix $H$ in section 4.1, and reflect it in the proof.

Reviewer **oyKA**:
> However, for non-quadratic loss functions the best pre-conditioner is not the inverse Hessian, and the proposed method will not learn the inverse Hessian. This is particularly obvious for non-convex loss functions, where the inverse Hessian is not even positive definite and Newton's method would not even converge to a minimum of the loss (see e.g. https://arxiv.org/abs/1406.2572)

Good catch, we missed the non-PSD case in our theory, and have now added it in. Our discussion originally required that $H$ is PSD to satisfy the stronger assumption that the approximation error $A = I-G(θ_t)H$ has spectral norm less than 1, since $||Ax||_2>||x||_2$ for any eigenvector $x$ of $H$ with negative eigenvalue. We can produce a further analysis when $H$ has some direction of negative curvature, if we restrict $G(\theta)$ to be PSD. In this case, $G(\theta)H$ must have a negative eigenvalue, implying that $A=I-G(\theta)H$ has an eigenvalue greater than 1, with corresponding eigenvector $y$. Since we then have $-G(θ_t)Hy = \lambda y$ for $\lambda > 0$ and $G(θ_t)$ is PSD, left multiplying both sides by $y^TG(θ_t)^{-1}$ further shows that $y^THy < 0$. This means that the error vector $b$ blows up exponentially in a direction $y$ which decreases the loss, and LODO will repel saddle points.

We have included this analysis of LODO in the non-PSD setting at the end of Section 4.1; along with a proof that $G(\theta)H$ has a negative eigenvalue in the Section B of the Appendix.

---

### Author Response · Authors · 2022-12-08
**We hope to engage in constructive discussion before the review period ends**

Dear Reviewers,

We would like to engage in constructive discussion before the end of the review period, since we believe we have addressed all of your concerns. We would highly appreciate any feedback you may have.

Best,

The Authors

---

### Decision · Program_Chairs · 2023-01-20

**Decision:**

Reject

**Justification For Why Not Higher Score:**

The paper needs more work to improve the clarity and comparison with existing related work.

**Justification For Why Not Lower Score:**

N/A

**Metareview: Summary, Strengths And Weaknesses:**

The paper proposes to learn a preconditioner that would approximate the inverse of the hessian thus resulting in faster training without having to explicitly compute the expensive hessian. This is achieved by jointly optimizing the objective in the iterates and preconditioner and performing a 1-step unrolled optimization. The paper then studies the convergence of the algorithm for a quadratic strongly convex objective and shows that the preconditioner converges towards the inverse of the Hessian. In addition, the authors propose a computationally cheap parameterization of the pre-conditioner which provably still retains good approximation capabilities. The method is tested on simulated and MNIST image generation experiments showing promising performance.
Given that the paper was borderline, there have been long discussions going on between the reviewers and AC after which a consensus was reached that the paper is currently not ready for publication to ICLR and would benefit from additional work to improve its clarity and comparisons with existing methods in the literature.


Here are some directions of improvement:

Reviewer oyKA raised concerns about the lack of empirical comparisons with methods such as natural gradient, etc which acts as pre-conditioners as well. Reviewer XFBK agrees that the comparisons are underwhelming. The paper could benefit from additional experiments to illustrate the gain of learning such pre-conditioners compared to developing faster approximations to better-understood conditioners.

In addition, oyKA pointed to a limited novelty since one-step unrolling of the inner loop optimization has been previously used to learn pre-conditioners. It appears that the main effective novelty is the use of momentum along with unrolling. Currently, the theoretical analysis does not account for the effect of momentum, which seems to be a key component for the improved results.

As pointed out by Reviewer w1wa, the paper lacks clarity, especially when presenting the theoretical contributions which make the reading hard to follow. Despite the improvements made by the authors during the rebuttal, the presentation of the approximation results is currently hard to parse and needs more context to introduce the required concepts.



**Summary Of Ac-Reviewer Meeting:**

There was a first discussion through the open review plateform with reviewer oyKA who provided detailed comments about the rebuttal.

After that, there was a meeting with the two other reviewers in which we discussed the paper as well as the comments of oyKA. The reviewers overall agreed with oyKA's comments, including myself. For this reason, I canceled the meeting with oyKA which was scheduled for a later time.

The discussion is summarized below:

Reviewer oyKA raised concerns about the lack of empirical comparisons with methods such as natural gradient, etc which acts as pre-conditioners as well. Reviewer XFBK agrees that the comparisons are underwhelming. The paper could benefit from additional experiments to illustrate the gain of learning such pre-conditioners compared to developing faster approximations to better-understood conditioners.

In addition, oyKA pointed to a limited novelty since one-step unrolling of the inner loop optimization has been previously used to learn pre-conditioners. It appears that the main effective novelty is the use of momentum along with unrolling. Currently, the theoretical analysis does not account for the effect of momentum, which seems to be a key component for the improved results.

As pointed out by Reviewer w1wa, the paper lacks clarity, especially when presenting the theoretical contributions which make the reading hard to follow. Despite the improvements made by the authors during the rebuttal, the presentation of the approximation results is currently hard to parse and needs more context to introduce the required concepts.